# PIPA: Preference Alignment as Prior-Informed Statistical Estimation

Junbo Li[1]   Zhangyang Wang[1]   Qiang Liu[1]

## Abstract

Offline preference alignment for language models such as Direct Preference Optimization (DPO) is favored for its effectiveness and simplicity, eliminating the need for costly reinforcement learning. Various offline algorithms have been developed for different data settings, yet they lack a unified understanding. In this study, we introduce Pior-Informed Preference Alignment (PIPA), a unified, RL-free probabilistic framework that formulates language model preference alignment as a Maximum Likelihood Estimation (MLE) problem with prior constraints. This method effectively accommodates both paired and unpaired data, as well as answer and step-level annotations. We illustrate that DPO and KTO are special cases with different prior constraints within our framework. By integrating different types of prior information, we developed two variations of PIPA: PIPA-M and PIPA-N. Both algorithms demonstrate a $3 \sim 10\%$ performance enhancement on the GSM8K and MATH benchmarks across all configurations, achieving these gains without additional training or computational costs compared to existing algorithms.

## 1. Introduction

Pre-training large language models (LLMs) from scratch on trillions of text tokens allows for accurate prediction next tokens in natural language (Achiam et al., 2023; Dubey et al., 2024; Liu et al., 2024b). Following this, alignment, achieved through fine-tuning on smaller, high-quality datasets designed for specific tasks, becomes critical for enabling the model to develop specialized skills, such as engaging in conversation (Ouyang et al., 2022), math reasoning (Shao et al., 2024; Yang et al., 2024), coding (Zhu et al., 2024),

web agent (Qin et al., 2025), and more. The fundamental approach to alignment involves supervised fine-tuning (SFT) on the target domain, which essentially maximizes the likelihood of predicting the next token. However, numerous empirical studies have shown that simple SFT on preferred samples is inadequate for attaining optimal performance (Shao et al., 2024; Ouyang et al., 2022).

Moving beyond basic imitation learning in SFT, it is suggested to learn from both positive and negative samples. Sample quality can be measured by training reward models to capture general preferences (Dong et al., 2024) or leveraging accurate rule-based rewards (Guo et al., 2025) for specific tasks like math and coding. By treating the autoregressive generation of LLMs as a Markov decision process (MDP), traditional reinforcement learning (RL) algorithms can be effectively applied, such as PPO (Ouyang et al., 2022), SAC (Liu et al., 2024c), REINFORCE (Ahmadian et al., 2024), etc.

While online RL-based methods deliver strong performance, they face challenges such as high training costs, instability, and the need for a strong base model as the initial policy. As a result, offline algorithms like direct preference optimization (DPO) (Rafailov et al., 2024) are often preferred, thanks to their effectiveness and simplicity, particularly when high-quality datasets are accessible. The original DPO algorithm has several limitations. It relies on paired preference data, which is not essential for tasks with ground truth such as math and coding. Additionally, it is unable to accommodate step-level annotations. Furthermore, it treats all tokens equally, lacking token-level credit assignment. To address these issues, a series of approaches have been developed, such as Kahneman-Tversky Optimization (KTO) (Ethayarajh et al., 2024) for unpaired data, Step-DPO (Lai et al., 2024; Lu et al., 2024) and Step-KTO (Lin et al., 2025) for step-level annotations, and RTO (Zhong et al., 2024), TDPO (Zeng et al., 2024), and OREO (Wang et al., 2024) for fine-grained token-level DPO. However, these methods are designed from specific perspectives, each addressing only particular challenges, and they lack a unified understanding to integrate their solutions.

In this work, we introduce a unified framework designed to address all the aforementioned challenges in offline ap-

[1]The University of Texas at Austin, US. Correspondence to: Qiang Liu <lqiang@cs.utexas.edu>.

*Proceedings of the $42^{nd}$ International Conference on Machine Learning*, Vancouver, Canada. PMLR 267, 2025. Copyright 2025 by the author(s).

proaches. Rather than framing the alignment problem within offline RL, we reformulate it as a maximum likelihood estimation (MLE) problem with prior constraints, operating within a purely probabilistic framework called Prior-Informed Preference Alignment (PIPA). From a statistical estimation perspective, we analyze the suboptimality of supervised fine-tuning (SFT). We demonstrate that both the original DPO and KTO algorithms can be interpreted as special cases within our framework, differing in the prior information they incorporate and the loss used. Building on the PIPA framework, we propose two variants, PIPA-M and PIPA-N, that incorporate prior information in different fashions. The probabilistic formulation naturally accommodates unpaired data and extends to step-level annotations. Our PIPA functions as a versatile plug-in loss design that seamlessly integrates with any (iterative) data generation pipeline in existing alignment framework. Furthermore, we show that PIPA training effectively learns token-level credit assignment, yielding precise per-token value estimations that may enable search during test-time inference.

Our contributions can be summarized as follows:

• We formulate preference alignment as a prior-informed conditional probability estimation problem that is RL-free and provides clear theoretical insight.

• Our approach does not need paired preference data, and seamlessly unifies both answer-wise and step-wise settings under a single, theoretically grounded framework.

• Compared to existing approaches such as DPO (Rafailov et al., 2024) and KTO (Ethayarajh et al., 2024), our algorithm achieves improved performance without additional computational overhead.

### 1.1. Related Work

**Learning from preference data**    RL has become a key framework for leveraging preference data for LLM alignment, with early methods like PPO (Schulman et al., 2017), which first trains a reward model on pairwise human feedback (Ouyang et al., 2022). Due to PPO's high training cost, direct policy optimization methods without online RL have been explored, integrating policy and reward learning into a single stage. Notable works include DPO (Rafailov et al., 2024), SLiC (Zhao et al., 2023), IPO (Azar et al., 2024), GPO (Tang et al., 2024), and SimPO (Meng et al., 2024). For fine-grained token-level optimization, DPO variants like TDPO (Zeng et al., 2024), TIS-DPO (Liu et al., 2024a), RTO (Zhong et al., 2024), and OREO (Wang et al., 2024) have been introduced. To address step-level annotation inspired by PRM (Lightman et al., 2023), methods such as Step-DPO (Lai et al., 2024), SCDPO (Lu et al., 2024), and SVPO (Chen et al., 2024b) have emerged. To relax pairwise data constraints, particularly for tasks with ground truth

like math and coding, KTO (Ethayarajh et al., 2024), Step-KTO (Lin et al., 2025), and OREO (Wang et al., 2024) have been proposed. Our PIPA framework addresses all these challenges within a unified paradigm, demonstrating that existing algorithms like DPO and KTO can be interpreted as special cases within our approach.

**Probabilistic alignment**    In addition to reward maximization, some research approaches alignment from a probabilistic perspective. (Abdolmaleki et al., 2024) decompose label likelihood into a target distribution and a hidden distribution, solving it using the EM algorithm. Other works leverage importance sampling to train a policy parameterized by an energy-based model that aligns with the target distribution including DPG (Parshakova et al., 2019), GDC (Khalifa et al., 2020), GDC++ (Korbak et al., 2022), BRAIn (Pandey et al., 2024). (Dumoulin et al., 2023) uses a density estimation formulation and recovers DPO. Unlike these methods, our PIPA framework maximizes label likelihood while enforcing prior constraints by transforming it into the target distribution using Bayes' Theorem. We directly learn the distributions without relying on complex sampling and estimation procedures. PIPA is scalable, incurs no additional training cost, and remains flexible across any preference data.

## 2. PIPA 🪕

We introduce the prior-informed preference alignment (PIPA) framework in Section 2.1, followed by the first version, PIPA-M, in Section 2.2. Next, we explore its connection to DPO (Rafailov et al., 2024) and KTO (Ethayarajh et al., 2024) in Section 2.3. Drawing inspiration from DPO and KTO, we develop the second version, PIPA-N, detailed in Section 2.4. In Section 2.5, we extend PIPA-M and PIPA-N naturally to incorporate step-level annotations. Finally, Section 2.6 presents the refined algorithms for PIPA-M and PIPA-N and compares them with prior methods like KTO.

### 2.1. Prior-Informed Preference Alignment

**Problem**    We define the preference alignment problem as probabilistic estimation. Assume that we are given a preference dataset

$$\{x^i, y^i, c^i\}_{i=1}^N \sim p^{\texttt{data}},$$

where $x^i$ is the instruction input, $y^i$ is the answer, and $c^i \in \{0, 1\}$ represents the preference or correctness of the answer. We are interested in predicting $y$ given $x$ in the correct case $(c = 1)$. This amounts to estimating conditional probability:

$$p(y \mid x, c = 1).$$

The canonical approach for estimating $p(y \mid x, c = 1)$ is, of course, the maximum likelihood estimation (MLE), which

yields supervised finetuning (SFT) on the positive examples with $c = 1$.

However, SFT only uses the positive samples, rendering the negative samples with $c = 0$ unusable. Preference alignment methods, on the other hand, aims to use both positive and negative data to get better estimation. But how is this possible while adhering to statistical principles, given that MLE is statistically optimal and $p(y \mid x, c = 1)$, by definition, involves only the positive data ($c = 1$)?

The idea is that the estimation should incorporate important prior information involving the negative data, thereby introducing a "**coupling**" between the estimations of the positive and negative probabilities, $p(y \mid x, c = 1)$ and $p(y \mid x, c = 0)$. Generally, this prior-informed estimation can be formulated as a constrained optimization problem that minimizes a loss on data fitness subject to a prior information constraint. In this work, we always set the data loss to be the log-likelihood of the label $c$:

$$\max_{\theta} \mathbb{E}_{p^{\text{data}}} \log p^{\theta}(c \mid x, y) \quad s.t. \quad p^{\theta} \in \texttt{PriorInfo.} \quad (1)$$

By assuming different prior information, we can derive various algorithms in a principled manner, with a transparent understanding of the underlying priors and preferences. The problem formulation in (1) naturally does MLE for each sample $(x, y, c)$ without the need of paired data as in DPO (Rafailov et al., 2024). If only $N$ pairs are available, we can decouple them into $2N$ samples as in KTO (Ethayarajh et al., 2024).

## 2.2. PIPA-M: Enforcing Prior on Marginal Distribution

We first consider a straightforward case when we know that the marginal prediction $p(y \mid x)$ should match a prior distribution $p^{\texttt{prior}}(y \mid x)$ defined by the pretrained LLM. Because the marginal distribution is a sum of the positive and negative probabilities, that is,

$$p(y \mid x) = p(y \mid x, c = 1)p(1|x) + p(y \mid x, c = 0)p(0|x),$$

where we abbreviate $p(c = i|x)$ as $p(i|x)$. The estimation of the positive and negative probabilities are coupled.

In this case, the estimation problem is in principle formulated as the constrained maximum likelihood problem (**PIPA-M**):

$$\max_{\theta} \quad \mathbb{E}_{p^{\text{data}}} \log p^{\theta}(c \mid x, y). \quad (2)$$
$$s.t. \quad p^{\theta}(y \mid x) = p^{\texttt{prior}}(y \mid x), \quad \forall x, y.$$

This is by definition the best way statistically to estimate $p$ provided the data information $p^{\texttt{data}}$ and the prior constraint $p^{\texttt{prior}}$.

**Parameterization of PIPA-M** Recall that our final goal is to estimate $p^{\theta}(y \mid x, c = 1)$, which is expected to be parameterized with an autoregressive transformer model. Hence, we are going to parameterize the target $p^{\theta}(c \mid x, y)$ using $p^{\theta}(y \mid x, c = 1)$ and constraint $p^{\theta}(y \mid x)$. This can be obtained by Bayes' rule:

$$\max_{\theta} \mathbb{E}_{p^{\text{data}}} p^{\theta}(c \mid x, y) = \mathbb{E}_{p^{\text{data}}} \frac{p^{\theta}(y \mid x, c)p^{\theta}(c \mid x)}{p^{\theta}(y \mid x)}, \quad (3)$$

which includes the two terms we are interested with an additional term $p^{\theta}(c \mid x)$. We set the two terms in the numerator to be learnable and the denominator to be fixed by prior constraint. Denote

$$p^{\theta}(y \mid x, c = 1) = f^{\theta}(y \mid x), \quad p^{\theta}(c = 1 \mid x) = g^{\theta}(x) \quad (4)$$

to be two learnable networks. The likelihood for positive and negative samples are:

$$p^{\theta}(c = 1 \mid x, y) = \frac{f^{\theta}(y \mid x)g^{\theta}(x)}{p^{\texttt{prior}}(y \mid x)}, \quad (5)$$
$$p^{\theta}(c = 0 \mid x, y) = 1 - p^{\theta}(c = 1 \mid x, y).$$

Therefore, we reformulate **PIPA-M** (2) as an unconstrained problem by directly maximizing the log-likelihood $\log p(c \mid x, y)$ via (5). The resulting loss is similar to KTO (Ethayarajh et al., 2024) that does not need preference pairs, but it differs notably in the loss formulation. A detailed comparison is provided in the next section.

In addition, in PIPA-M we notice that:

- Since $p^{\theta}(y, c \mid x) = p^{\theta}(c \mid x, y)p^{\theta}(y \mid x)$, PIPA-M problem (2) is equivalent to directly maximizing the joint probability $p^{\theta}(y, c \mid x)$ with $p^{\theta}(y \mid x)$ being a fixed prior.

- It's possible that the parameterization with $f^{\theta}$ and $g^{\theta}$ makes $p^{\theta}(c = 1 \mid x, y)$ large than 1. Theoretically, we can first use a network $g_0^{\theta}$ and than set $g^{\theta}(x) = \min(g_0^{\theta}(x), p^{\texttt{prior}}(y \mid x)/f^{\theta}(y \mid x))$ to ensure the term to be well-defined. In practice, we observe that such cases are rare and we just apply clipping outsides of $\frac{f^{\theta}(y|x)g^{\theta}(x)}{p^{\texttt{prior}}(y|x)}$ to make it smaller than 1.

## 2.3. DPO and KTO: Prior-Informed Views

We analyze existing methods, such as DPO and KTO, through the lens of the prior-informed estimation framework (1). Our analysis demonstrates that both DPO and KTO can be interpreted as enforcing **prior assumptions on the negative probability**, $p^{\theta}(y \mid x, c = 0)$, **rather than the marginal probability**, $p(y \mid x)$. However, these methods differ in their choice of loss functions, the prior assumptions made about $p(c \mid x)$, and the parameterization employed.

---

**Algorithm 1** PIPA: Prior-Informed Preference Alignment

---

1: **Input:** A dataset $\{(x, y, c)\}$ of questions $x$, answers $y$, and preference $c$, with $c \in \{0, 1\}^T$; a fixed prior model $p^{\texttt{prior}}$, and trainable models $f^\theta$ and $g^\theta$ initialized from an SFT model $f_0$. Choice: either PIPA-M or PIPA-N.

2: **for** each batch **do**

3:     **PIPA-M:** Compute

$$F^\theta(x, y_{\leq t}) := \frac{f^\theta(y_t \mid x, y_{<t}) \, g^\theta(x, y_{<t})}{p^{\texttt{prior}}(y_t \mid x, y_{<t})}.$$

    **PIPA-N:** Compute

$$F^\theta(x, y_{\leq t}) := \tau\left(\frac{f^\theta(y_t \mid x, y_{<t}) \, g^\theta(x, y_{<t})}{p^{\texttt{prior}}(y_t \mid x, y_{<t}) \, (1 - g^\theta(x, y_{<t}))}\right),$$

where $\tau(x) := \frac{x}{x+1}$.

4:     Minimize the loss:

$$L(x, y, c) = -\sum_{t:c_t=1} \log F^\theta(x, y_{\leq t}) - \sum_{t:c_t=0} \log\left(1 - F^\theta(x, y_{\leq t})\right).$$

5: **end for**

---

**DPO** Direct preference optimization (DPO) and related methods are often cast as estimating the log density ratio $\log(p(y \mid x, c = 1)/p(y \mid x, c = 0))$ of the positive and negative generation probabilities (Dumoulin et al., 2023). Related, it may not be of suprise that these models make the implicit assumption on the negative (reference) probability $p(y \mid x, c = 0) = p^{\texttt{prior}}(y \mid x)$. In particular, DPO can be formulated as

$$\max_\theta L_{\texttt{DPO}}(p^\theta, \ p^{\texttt{data}}) \qquad (6)$$
$$s.t. \ \ p^\theta(y \mid x, c = 0) = p^{\texttt{prior}}(y \mid x),$$
$$p^\theta(c = 1 \mid x) = p^\theta(c = 0 \mid x) = \frac{1}{2}, \quad \forall x, y.$$

where the loss $L_{\texttt{DPO}}$ is a special pairwise comparison loss related to Bradley–Terry model provided paired data $\{x, y^+, y^-\}$, of both positive answer $y^+ \sim p(y \mid x, c = 1)$ and negative answer $y^- \sim p(y \mid x, c = 0)$ for each $x$; the assumption of $p^\theta(c = 1 \mid x) = 0.5$ is due to the balanced sampling of positive and negative weights. See Appendix A.1 for more discussion for $L_{\texttt{DPO}}$ and proof of equivalence.

**KTO** One key limitation of DPO is that it requires to use paired data. KTO has been proposed as an approach that relaxes the requirement. In the prior-informed framework, it can be viewed as solving:

$$\max_\theta \mathbb{E}_{p^{\texttt{data}}}\left[p^\theta(c|x, y)\right]$$
$$s.t. \ \ p^\theta(y \mid x, c = 0) = p^{\texttt{prior}}(y \mid x)$$
$$\log \frac{p^\theta(c = 0 \mid x)}{p^\theta(c = 1 \mid x)} = z^\theta(x), \qquad \forall x, y.$$

where it changes the loss function to the standard conditional likelihood without log, which holds for unpaired data[1]. In

---
[1] In the KTO paper, an importance weight is placed on the positive and negative data ($\lambda_D, \lambda_U$ in their notation), but the default

addition, it makes a particular assumption on the class ratio $p^\theta(c = 0 \mid x)/p^\theta(c = 1 \mid x)$, which is consistent with the fact that the class percentage is no longer guaranteed to be balanced without paired data. In particular, KTO uses

$$z^\theta(x) = \texttt{KL}(p^\theta(y \mid x, c = 1) \,||\, p^{\texttt{prior}}(y \mid x)).$$

Here $z^\theta$ depends on parameter $\theta$, but the gradient is stopped through $z^\theta$ in the KTO algorithm. In practice, $z^\theta$ is estimated with empirical samples in each batch. Details and proof are shown in Appendix A.2.

### 2.4. PIPA-N: Enforcing Prior on Negative Condition Distribution

Knowing the prior informed formulation of DPO and KTO, we can propose changes to make them simpler and more natural. One option is to keep the conditional log-likelihood loss function of KTO, but seeks to learn $p^\theta(c = 1|x)$ as a neural network $g^\theta(x)$ from data, rather than making the heuristic assumption. This yields

$$\max_\theta \mathbb{E}_{p^{\texttt{data}}}\left[\log p^\theta(c \mid x, y)\right]$$
$$s.t. \ \ p^\theta(y \mid x, c = 0) = p^{\texttt{prior}}(y \mid x). \quad \forall x, y. \qquad (7)$$

The only difference with PIPA-M (2) is in the prior constraint. We call this **PIPA-N**, for it places prior on the negative probability. As in PIPA-M, we apply Bayes' rule to $p^\theta(c \mid x, y)$, but additionally expand the denominator $p^\theta(y \mid x)$ using:

$$p^\theta(y \mid x) = p^\theta(y \mid x, c = 1)p^\theta(c = 1 \mid x)$$
$$+ p^\theta(y \mid x, c = 0)p^\theta(c = 0 \mid x)$$

This allows us to incorporate prior information on $p^\theta(y \mid x, c = 0)$. With the parameterizations $p^\theta(y \mid x, c = 1) =$

---
settings in the code is balanced weight ($\lambda_D = \lambda_U = 1$).

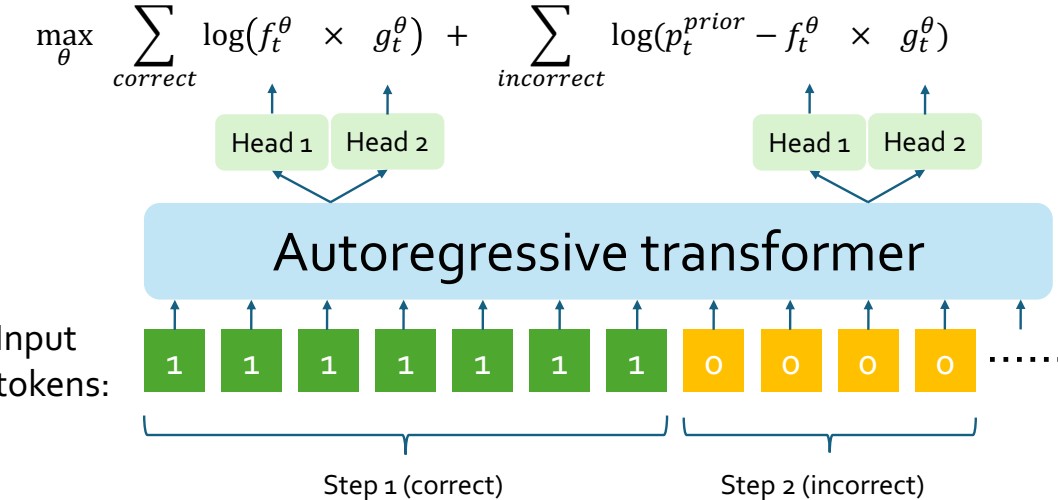

$$\max_{\theta} \sum_{correct} \log(f_t^{\theta} \times g_t^{\theta}) + \sum_{incorrect} \log(p_t^{prior} - f_t^{\theta} \times g_t^{\theta})$$

Head 1   Head 2      Head 1   Head 2

Autoregressive transformer

Input tokens:

1   1   1   1   1   1   1    0   0   0   0   ......

Step 1 (correct)      Step 2 (incorrect)

*Figure 1.* The figure illustrates PIPA-M, with PIPA-N following a different loss function as outlined in Algorithm 1. We denote $f_t^{\theta} := f^{\theta}(y_t \mid x, y_{<t}) = p^{\theta}(y_t \mid x, y_{<t})$ to be the target next-token-prediction probability, $g_t^{\theta} := g^{\theta}(x, y_{<t}) = p^{\theta}(c_t = 1 \mid x, y_{<t})$ to be the value model, which differ only in their output heads. And $p_t^{prior} := p^{prior}(y_t \mid x, y_{<t})$ is another frozen language model. When only answer-level annotations are available, the objective will contain only one of the correct and incorrect parts.

$f^{\theta}(y \mid x)$ and $p^{\theta}(c = 1 \mid x) = g^{\theta}(x)$, we obtain:

$$p^{\theta}(c = 1 \mid x, y) = \tau \left( \frac{f^{\theta}(y \mid x)g^{\theta}(x)}{p^{prior}(y \mid x)(1 - g^{\theta}(x))} \right),$$

$$p^{\theta}(c = 0 \mid x, y) = 1 - p^{\theta}(c = 1 \mid x, y),$$

where $\tau(x) = x/(x + 1)$. This allows us to reduce the problem to an unconstrained optimization of maximizing $\mathbb{E}_{p^{\text{data}}} \left[ \log p^{\theta}(c \mid x, y) \right]$.

## 2.5. Extension to Step-level Settings

The introduction above is about the answer-level setting, which uses a single label for an entire solution—even if multiple steps may vary in correctness. In contrast, the step-level setting assigns separate labels to each step. The advantage of our probabilistic framework is that it seamlessly adapts to a step-level setting for both PIPA-M and PIPA-N algorithms. The core idea is intuitive. We use token-level labels rather than answer-level labels, followed by decomposing the joint probability autoregressively.

**Problem formulation** In the step-level setting, we decompose answer $y$ and label $c$ to tokens. Specifically, for each data $(x, y, c)$ with $k$ steps and $T$ tokens in the answer, we have $y = (y_1, \cdots, y_T)$ and $c = (c_1, \cdots, c_T) \in \{0, 1\}^T$, where $c_t$ is 1 if the corresponding step is correct otherwise 0. Notice that if only answer-level annotation is available, we

can still define $c_1 = \cdots = c_T \in \{0, 1\}$. For any $1 \le t \le T$, denote $y_{\le t} := (y_1, \cdots, y_t)$ and the same for $c_{\le t}$. Figure 2 presents a visualization comparing token-level representation with the previous sequence-level representation.

| Representation | Annotation | Label |
| --- | --- | --- |
| Sequence | Answer | 0 |
| Token | Answer | 0 0 0 0 0 0 |
| Token | Step | 1 1 1 1 0 0 |

*Figure 2.* We show a visualization for the label of a negative answer under different circumstances.

**Parameterization** Same as the answer-level setting, the objective is still (2) for PIPA-M or (7) for PIPA-N. We factorize $p^{\theta}(c \mid x, y)$ in an autoregressive manner for $c$. Since $c_t$ can be determined by $(x, y_{\le t})$, so $c_t$ is **conditionally independent** of both $y_{>t}$ and $c_{<t}$ given $(x, y_{\le t})$. We have

$$p(c \mid x, y) = \prod_t p(c_t \mid x, y, c_{<t}) = \prod_t p(c_t \mid x, y_{\le t}).$$

By Bayes' Theorem, for each $t$ we have:

$$p(c_t \mid x, y_{\le t}) = \frac{p(y_t \mid x, y_{<t}, c_t)p(c_t \mid x, y_{<t})}{p(y_t \mid x_t, y_{<t})}.$$

Now similar to the answer-level PIPA, we introduce neural networks to parameterize $f^\theta(y_t \mid x, y_{<t}) := p^\theta(y_t \mid x, y_{<t}, c_t = 1)$ and $g^\theta(x, y_{<t}) := p^\theta(c_t = 1 \mid x, y_{<t})$.

Then we solve the same unconstrained optimization problem as answer-level setting, *i.e.,* maximizing $\mathbb{E}_{p^{\texttt{data}}}[\log p^\theta(c \mid x, y)]$. See Algorithm 1 for details.

## 2.6. Practical Implementation

As shown in the formulation of Section 2.5, we always use PIPA-M and PIPA-N with token-level label representation in practice which provides fine-grained information.

We have three models in total: $f^\theta, g^\theta, p^{\texttt{prior}}$. In practice, $f^\theta$ is the target language model initialized from the one obtained after Supervised Fine-Tuning (SFT) and will be used for inference after training. $g^\theta$ shares exactly the same base model with $f^\theta$ differing only in the output head. $p^{\texttt{prior}}$ is also a language model but frozen. We show our PIPA-M and PIPA-N in Algorithm 1, and a figure illustration in Figure 1. For the negative samples in PIPA-M, we apply a clipping function to constrain the term $f^\theta g^\theta / p^{\texttt{prior}}$ within $[0, 1 - \varepsilon]$, where $\varepsilon = 10^{-6}$.

**Comparison with KTO** Although it stems from a completely different derivation, KTO (Ethayarajh et al., 2024)—which also targets pair-free alignment—is the closest prior work to PIPA in terms of its algorithm. A detailed analysis of their differences and PIPA's advantages can be found in Appendix A.2.1.

**Credit assignment** A key issue with traditional DPO and KTO loss is that all tokens receive equal treatment, since only the answer-level reward is considered. PIPA offers a natural way to weight $f^\theta(y_t \mid x, y_{<t})$ differently by jointly optimizing it with $g^\theta(x, y_{<t}) = p^\theta(c_t = 1 \mid x, y_{<t})$. The optimized $g^\theta$, with its clear probabilistic interpretation, can be viewed as a value function and may be used for inference-time search in future work. We present its learning trajectory in Section 3.3.1.

**Compatibility and flexibility** PIPA can be applied whenever answer- or step-level annotations are available, requiring no additional training stage. These step-level annotations can be derived via MCTS (Chen et al., 2024a; Zhang et al., 2024b; Guan et al., 2025) or by LLM-as-a-judge (Lai et al., 2024; Lin et al., 2025). Furthermore, PIPA easily generalizes to an iterative version, similar to other works (Xiong et al., 2024; Pang et al., 2024; Wang et al., 2024). In this paper, we focus on statistical estimation using a static offline dataset, leaving the online version for future exploration.

## 3. Experiments

### 3.1. Settings

Our PIPA framework is capable of handling scenarios both with and without preference pairs, as well as with or without step-level annotations. Consequently, we evaluate it across four distinct experimental setups determined by $(\text{pair}, \text{unpair}) \times (\text{answer}, \text{step})$. In this work, we primarily focus on math reasoning tasks, as they serve as a strong testbed for the scenarios under consideration. We leave systematic exploration of general tasks for future work.

**Baseline algorithms** For paired preference data, we use DPO (Rafailov et al., 2024) and its variant IPO (Azar et al., 2024), and KTO (Ethayarajh et al., 2024) as baselines. Both KTO and PIPA decouple the paired data. For data without preference pairs, we compare PIPA with KTO. In answer-wise settings, we benchmark PIPA against DPO, IPO, and KTO. In step-wise settings, we compare PIPA with Step-DPO (Lai et al., 2024) and Step-KTO (Lin et al., 2025). The original Step-DPO and Step-KTO methods involve additional data generation phase. For a fair comparison on an offline dataset, we extract only their loss functions. See Appendix B for detailed descriptions of the baseline algorithms.

**Data** We use the unpaired preference dataset for math reasoning released by AlphaMath (Chen et al., 2024a), which includes training problems from GSM8K (Cobbe et al., 2021) and MATH (Hendrycks et al., 2021) with both CoT (Wei et al., 2022) and TIR (Gou et al., 2023)-style solutions, along with step-level label annotations. There are more incorrect answers in this dataset, so we construct the paired subset by matching each correct solution to a corresponding incorrect solution from the same problem and discarding the remaining incorrect solutions. We use the entire original dataset for the unpaired setting.

For the answer-level setting, we only keep the final label for the answer. In the step-level setting, steps of a correct answer are always correct. For incorrect answer, we label steps whose $Q$ values fall within $[-1, 0.5)$ as incorrect, and those in $[0.5, 1]$ as correct with a threshold $0.5$. Instead of a threshold $0$, the intuition of this shifted threshold is that it's better to be conservative for the correct steps in the wrong answer. Despite being correct, some steps may still contribute to an incorrect overall analysis. Therefore, minimizing the likelihood of such steps is also crucial. The efficacy of this choice is further explored in the ablation study of Section 3.3.2.

For our evaluation, we use the standard GSM8K and MATH benchmarks. We adopt the MARIO evaluation toolkit (Zhang et al., 2024a), configuring the beam search width and number of generated samples, *i.e.*, $(B_1, B_2)$ in

their notation, to be $(3, 1)$ for GSM8K and $(1, 1)$ for MATH.

**Model** The AlphaMath dataset is generated using Deepseek-based models, so we use Deepseek-Math-7B-Instruct (Shao et al., 2024) as the pre-trained model for self-alignment. Additionally, to evaluate generalization capabilities, we test Qwen2.5-Math-7B-Instruct (Yang et al., 2024) as the base model on the same dataset. For PIPA, we set the head of $g^\theta$ to be a two-layer MLP followed by a Sigmoid function, with hidden dimension 4096 same as the base model.

**Training** All experiments are conducted based on Open-RLHF (Hu et al., 2024). For all training, we use LoRA (Hu et al., 2021) with rank 64 and $\alpha = 16$. All alignment algorithms are conducted for 1 epoch after the SFT stage. Denote $bs$ to be the batch size and $lr$ to be the learning rate. we do grid search for $lr \in \{5 \times 10^{-7}, 5 \times 10^{-6}, 5 \times 10^{-5}\}$ for all experiments and present the best one.

- **SFT** Before all alignment algorithms, we first fine-tune the pre-trained Deepseek and Qwen models on the positive samples for 3 epochs with $bs = 1024$ and $lr = 4 \times 10^{-5}$. The model obtained after SFT is then used as the initilization for the target model $f^\theta$ in alignment procedures, as well as the fixed reference model in DPO and KTO. Furthermore, to avoid extra computation, this same model serves as the prior in both PIPA-M and PIPA-N, ensuring that PIPA does not require an additional training phase compared to DPO and KTO.

- **DPO** For DPO-based algorithms including DPO, IPO, Step-DPO, we train 1 epoch after the SFT stage, with $bs = 256, lr = 5 \times 10^{-7}$ and $\beta = 0.1$.

- **KTO** For KTO, we set $lr = 5 \times 10^{-5}$ for Deepseek model and $lr = 5 \times 10^{-7}$ for Qwen model. For both, $bs = 256, \beta = 0.1$. Step-KTO shares exactly the same recipe with KTO.

- **PIPA** We set $bs = 256$ for all four settings, $lr = 5 \times 10^{-5}$ for Deepseek and $5 \times 10^{-7}$ for Qwen. All settings are the same as KTO and Step-KTO, without additional hyperparameters to be tuned.

## 3.2. Main Results

We show our main results in Table 1. We can see that for all four settings and two models, PIPA achieves the best performance without additional computation cost.

## 3.3. Additional Analysis and Ablation Studies

We conduct a more detailed analysis from two perspectives. Section 3.3.1 delves into further studies on our PIPA it-

self. Section 3.3.2 examines the step-level and answer-level settings.

### 3.3.1. ALGORITHMS

**PIPA-M vs. PIPA-N** PIPA-M and PIPA-N are two versions of our framework that incorporate distinct prior constraints. As shown in Table 1, neither variant consistently outperforms the other. Notably, PIPA-N tends to perform better with the Deepseek model, while PIPA-M shows superior results with the Qwen model. This may suggest that PIPA-N is better suited for self-alignment scenarios, while PIPA-M is more effective for alignment tasks where there is a distribution shift between the model and the data. In practice, we recommend experimenting with both variants to determine the optimal choice for your specific use case.

**Value model** $p(c_t \mid x, y_{<t})$ Our framework employs two components: the target model $p^\theta(y \mid x, c = 1)$ and a value model $p^\theta(c_t = 1 \mid x, y_{<t})$. These are jointly trained through optimization of their combined probabilistic objective, rather than being learned separately. Downstream task evaluations confirm that $p^\theta(y \mid x, c = 1)$ is well optimized. To assess the impact of the value model, we present the results of removing it in Table 2, where the performance decline highlights its importance. To examine the value model's learning behavior, we plot the training trajectory of $\prod_t \left( p^\theta(c_t \mid x, y_{<t}) \right)^{1/T}$ using Deepseek model in the step-wise setting for the first 300 steps in Figure 3. The implicit optimization process yields continuous improvement in likelihood

| GSM8K | MATH |
|-------|------|
| **78.24** | **51.82** |
| 73.54 | 47.92 |

*Table 2.* In the second row, $p^\theta(c_t \mid x, y_t)$ is fixed at 0.5.

estimation, with the model's predictions showing a steady increase from the initial random-guess baseline of 0.5. This empirical validation establishes a foundation for using the optimized value model $p^\theta(c_t = 1 \mid x, y_{<t})$ to implement search during inference, presenting a clear direction for future research.

**Prior choices** To ensure a fair comparison with baseline methods such as DPO and KTO, we utilize the same SFT model for initializing $p^\theta(y \mid x, c = 1)$ and setting priors. In our PIPA model, however, the priors should ideally be $p(y \mid x)$ for PIPA-M and $p(y \mid x, c = 0)$ for PIPA-N, which differ from the SFT model's $p^\theta(y|x, c = 1)$. To further investigate PIPA with accurate priors, we began with the released Deepseek model, training it on both positive and negative samples for three epochs to derive $p^\theta(y \mid x)$, and solely on negative samples for three epochs to obtain $p^\theta(y \mid x, c = 0)$. Unfortunately, these adjustments did not yield improvements. Results are shown in Table 3. This

| Data | Annotation | Algorithm | GSM8K | | MATH | |
|---|---|---|---|---|---|---|
| | | | DS | QW | DS | QW |
| Paired | Answer-wise | DPO (Rafailov et al., 2024) | 68.39 | 67.17 | 46.94 | 47.78 |
| | | IPO (Azar et al., 2024) | 69.14 | 72.33 | 46.94 | 49.96 |
| | | KTO (Ethayarajh et al., 2024) | 76.72 | 62.47 | 47.38 | 46.53 |
| | | **PIPA-M** | 79.08 | **73.77** | 50.82 | **51.60** |
| | | **PIPA-N** | **80.29** | 70.89 | **52.32** | 47.26 |
| | Step-wise | Step-DPO (Lai et al., 2024) | 68.54 | 66.11 | 46.96 | 48.38 |
| | | Step-KTO (Lin et al., 2025) | 75.44 | 62.47 | 47.38 | 45.64 |
| | | **PIPA-M** | **79.15** | **74.91** | 51.94 | **53.26** |
| | | **PIPA-N** | 78.70 | 73.84 | **52.54** | 49.06 |
| Unpaired | Answer-wise | KTO (Ethayarajh et al., 2024) | 76.04 | 64.44 | 46.72 | 47.08 |
| | | **PIPA-M** | 79.08 | **74.75** | 51.04 | **52.78** |
| | | **PIPA-N** | **80.97** | 74.22 | **52.22** | 52.00 |
| | Step-wise | Step-KTO (Lin et al., 2025) | 76.81 | 64.14 | 46.98 | 45.64 |
| | | **PIPA-M** | 78.24 | **74.22** | 51.82 | **53.10** |
| | | **PIPA-N** | **79.98** | 72.86 | **52.78** | 52.52 |

*Table 1.* Results on GSM8K and MATH. DS means Deepseek-based models, and QW means Qwen-based models.

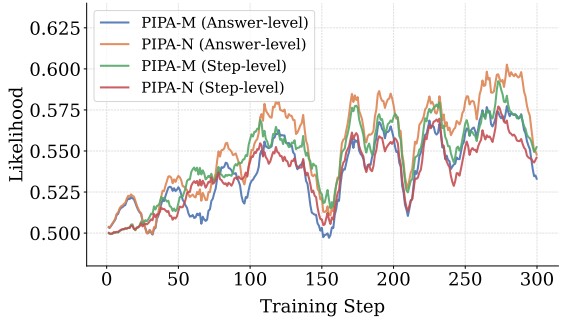

*Figure 3.* We plot the geometrically averaged likelihood $\left(\Pi_t p^\theta(c_t \mid x, y_{<t})\right)^{\frac{1}{T}}$ for PIPA-M and PIPA-N with answer-level annotation and step-level annotation respectively during training, showing a consistent increase.

may be due to the lack of training on marginal or negative samples in the initial pre-training and fine-tuning stages of model development, meaning that a few epochs of fine-tuning are insufficient to establish accurate priors for these distributions.

**Additional SFT loss** Previous studies (Pang et al., 2024; Dubey et al., 2024) have shown that incorporating an additional SFT loss in DPO enhances its stability. We extend it to the unpaired setting, applying it to our PIPA-M and KTO algorithms in the answer-wise case, with an SFT loss coefficient set at 1.0. As shown in Table 4, incorporating additional SFT loss provides more advantages for KTO compared to our PIPA, yet it remains less effective than our PIPA. This is because our PIPA is theoretically grounded for a general case and does not require further modifications

to the loss function.

| | GSM8K | MATH |
|---|---|---|
| PIPA-M | **78.24** | **51.82** |
| PIPA-M(T) | 77.86 | 50.60 |
| PIPA-N | **79.98** | **52.78** |
| PIPA-N(T) | 79.83 | 50.84 |

*Table 3.* PIPA with different priors. (T) denotes further fine-tuning of the SFT model on all or negative samples.

| | GSM8K | MATH |
|---|---|---|
| KTO | 76.04 | 46.72 |
| KTO+SFT | 76.27 | 47.96 |
| PIPA-M | **79.08** | **50.82** |
| PIPA-M+SFT | 78.24 | 50.12 |

*Table 4.* Effect of additional SFT loss on KTO and PIPA-M.

### 3.3.2. STEP-LEVEL SETTING

As our research is the first to systematically explore the performance of alignment algorithms across various settings, we provide an in-depth analysis of the step-level setting in this section. We aim to understand the advantages of step-level annotation and how it influences the training.

**Influence of step-level annotation** From Table 1, we observe that step-level annotation does not consistently improve performance when comparing answer-wise and step-wise annotation within the same algorithm and dataset. Specifically, step-level annotation proves beneficial for MATH but can sometimes negatively impact GSM8K. This finding aligns with previous studies (Chen et al., 2024a), suggesting that step-level annotation is more advantageous for challenging reasoning tasks like MATH but may be unnecessary or even harmful for simpler tasks like GSM8K.

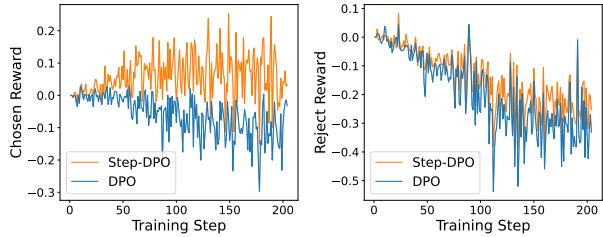

*Figure 4.* We present plots of $\log \frac{p^\theta(y|x,c=1)}{p^{\text{prior}}(y|x))}$ for both correct and incorrect samples, shown in the left and right figures respectively, for both the original DPO and Step-DPO. While the term for correct samples decreases as observed in previous studies, it increases in Step-DPO.

**Reward curve** As shown in previous works (Pang et al., 2024; Dubey et al., 2024; Razin et al., 2024; Liu et al., 2024d), one problem in DPO is that the implicit reward for positive samples $\log \frac{p^\theta(y|x,c=1)}{p^{\text{prior}}(y|x))}$ can also decrease during preference learning—undesirable, since this term is precisely what DPO aims to optimize. Their approach addresses this problem by adding extra SFT loss during DPO training. We have noticed similar patterns in our DPO experiments. Notably, we found that employing step-level annotation can effectively address this issue. This observation offers an alternative angle for tackling the problem in DPO, stemming from the absence of step-level annotation. It's possible that some steps in incorrect answers are actually correct and share similarities with the distribution of correct answers. Consequently, minimizing these correct steps in incorrect answers with the original answer-level DPO could also reduce the likelihood of correct answers. Our results highlight the importance of fine-grained, step-level annotation alignment from a new perspective.

**Threshold for positive steps** The step-level annotation, specifically the Q value obtained by MCTS in our AlphaMath dataset (Chen et al., 2024a), is presented in a continuous format ranging from $[-1, 1]$. The similar continuous format is employed in other annotation pipelines such as LLM-as-a-judge (Lai et al., 2024). In our main experiments, we employ a default threshold of 0.5 for labeling correct steps in incorrect answers. As analyzed in Figure 5, which evaluates the impact of varying threshold values, we observe that an intermediate threshold achieves optimal performance. This balance ensures cautious filtering of positive steps in negative

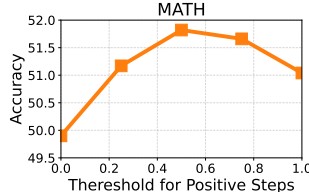

*Figure 5.* Accuracy on MATH with different threshold for the positive steps in wrong answers.

answers while retaining sufficient high-quality positive steps to maintain learning efficacy.

## 4. Conclusion

In this paper, we introduce Prior-Informed Preference Alignment (PIPA), a fully probabilistic framework grounded in statistical estimation. We analyze the limitations of pure SFT within this framework and demonstrate that DPO and KTO emerge as special cases with distinct prior constraints. We propose two variants, PIPA-M and PIPA-N, each incorporating different prior constraints. Through comprehensive evaluation across four distinct data settings, we systematically highlight PIPA's advantages over previous methods. Additionally, ablation studies reveal the optimal design of PIPA, while empirical analysis explores the impact of step-level annotation from multiple perspectives, leveraging PIPA's flexibility.

## Acknowledgment

We thank the anonymous reviewers for their helpful suggestions. Z. Wang is in part supported by NSF Award 2145346 (CAREER) and 2212176. This research has been supported by computing support on the Vista GPU Cluster through the Center for Generative AI (CGAI) and the Texas Advanced Computing Center (TACC) at the University of Texas at Austin.

## Impact Statement

This paper introduces a principled statistical framework for LLM alignment that allows us to both understand existing methods and deriving new efficient ones. Experimental results highlight its effectiveness in improving LLMs' mathematical reasoning, while its applicability extends to broader social implications, such as enhancing LLM safety, in line with previous research in the field.

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

# A. Detailed Discussion about Connection to DPO and KTO

## A.1. The DPO Loss

DPO uses a pairwise comparison loss on paired positive and negative data. Denote the pair data to be $\{x^i, y^{i^+}, y^{i^-}\}$, where $y^{i^+}$ is the chosen answer sampled from $p(y \mid x, c = 1)$ and $y^{i^-}$ sampled from $p(y \mid x, c = 0)$ is the rejected answer. Using our notation, the DPO objective (Rafailov et al., 2024) is

$$\max_{\theta} \mathbb{E}[\log \sigma \left( r^{\theta}(y^+, y^-, x) \right)], \tag{8}$$

where

$$r^{\theta}(y^+, y^-, x) = \log \frac{p^{\theta}(y^+ \mid x, c = 1)p^{\texttt{prior}}(y^- \mid x)}{p^{\theta}(y^- \mid x, c = 1)p^{\texttt{prior}}(y^+ \mid x)}.$$

From our perspective, DPO can be viewed as minimizing a pairwise comparison loss, subject to an prior assumption on the negative probability $p(y \mid x, c = 0)$, rather than the marginal probability $p(y \mid x)$. We show it in the following Theorem.

**Theorem A.1.** *Draw $(x, y, c)$ from a joint distribution $p$, for which $p(c = 1 \mid x) = 1/2$. Further, set the sample $y^c = y$, and draw an independent contrastive sample via $y^{\neg c} \sim p(\cdot \mid x, 1 - c)$. Then maximizing the original DPO objective (8) is equivalent to solving the following problem:*

$$\begin{aligned}
\max_{\theta} \quad & \mathbb{E}_p \left[ \log p^{\theta}(c \mid x, (y^c, y^{\neg c})) \right] \\
s.t. \quad & p^{\theta}(y \mid x, c = 0) = p^{\texttt{prior}}(y \mid x), \\
& p^{\theta}(c = 1 \mid x) = p^{\theta}(c = 0 \mid x) = 0.5, \forall x, y.
\end{aligned} \tag{9}$$

*Proof.* Using Bayes' Theorem, we have:

$$p^{\theta}(c \mid x, (y^c, y^{\neg c})) = \frac{p^{\theta}((y^c, y^{\neg c}) \mid x, c)p^{\theta}(c \mid x)}{p^{\theta}((y^c, y^{\neg c}) \mid x)}. \tag{10}$$

Denote

$$h^{\theta}(y^c, y^{\neg c}, x, c) := p^{\theta}(y^c \mid x, c)p^{\theta}(y^{\neg c} \mid x, 1 - c),$$

Define $(y^+, y^-) = \mathbb{1}_{\{c=1\}}(y^c, y^{\neg c}) + \mathbb{1}_{\{c=0\}}(y^{\neg c}, y^c)$. We have

$$\begin{aligned}
p^{\theta}(c \mid x, (y^c, y^{\neg c})) &= \frac{h^{\theta}(y^c, y^{\neg c}, x, c)p^{\theta}(c \mid x)}{h^{\theta}(y^c, y^{\neg c}, x, c)p^{\theta}(c \mid x) + h^{\theta}(y^c, y^{\neg c}, x, 1 - c)p^{\theta}(1 - c \mid x)} \\
&= \frac{p^{\theta}(y^c \mid x, c)p^{\theta}(y^{\neg c} \mid x, 1 - c)}{p^{\theta}(y^c \mid x, c)p^{\theta}(y^{\neg c} \mid x, 1 - c) + p^{\theta}(y^c \mid x, 1 - c)p^{\theta}(y^{\neg c} \mid x, c)} \\
&= \frac{p^{\theta}(y^+ \mid x, c = 1)p^{\texttt{prior}}(y^- \mid x)}{p^{\theta}(y^+ \mid x, c = 1)p^{\texttt{prior}}(y^- \mid x) + p^{\texttt{prior}}(y^+ \mid x)p^{\theta}(y^- \mid x, c = 1)}
\end{aligned} \tag{11}$$

On the other hand, notice that $\log \sigma(x) = -\log(1 + \exp(-x))$. So the original DPO loss is

$$\begin{aligned}
& \max_{\theta} \log \sigma \left( r^{\theta}(y^+, y^-, x) \right) \\
&= -\log(1 + \exp(-r^{\theta}(y^+, y^-, x))) \\
&= -\log \left( 1 + \frac{p^{\theta}(y^- \mid x, c = 1)p^{\texttt{prior}}(y^+ \mid x)}{p^{\theta}(y^+ \mid x, c = 1)p^{\texttt{prior}}(y^- \mid x)} \right).
\end{aligned}$$

From this, it's straightforward to see that DPO loss (8) is equivalent to applying $-\log()$ to (11). □

Therefore, Theorem A.1 shows that DPO is well recovered by our framework. In DPO, besides injecting prior for $p(y \mid x, c = 0)$ instead of $p(y \mid x)$, it has additional prior $p^{\theta}(c = 1 \mid x) = p^{\theta}(c = 0 \mid x)$. A direct idea is to remove this prior, and set $p^{\theta}(c = 1 \mid x)$ to be a learnable model for $x$ similar to PIPA-M.

## A.2. Connection to KTO

The KTO objective is

$$\max_\theta \mathbb{E}_{p^{\text{data}}} \quad [c\sigma\left(h_\theta(x,y) - z(x)\right) + (1-c)\sigma\left(-h_\theta(x,y) + z(x)\right)], \tag{12}$$

where

$$h_\theta(x,y) := \log\frac{p^\theta(y \mid x, c=1)}{p^{\text{prior}}(y \mid x)},$$

and

$$z(x) := \text{KL}(p^\theta(y \mid x, c=1)\|p^{\text{prior}}(y \mid x))).$$

We show the following equivalence.

**Theorem A.2.** *Maximizing the original KTO objective ([12](#)) is equivalent to solving the following problem:*

$$\max_\theta \mathbb{E}_{p^{\text{data}}}\left[p^\theta(c|x,y)\right] \tag{13}$$

$$\text{s.t.} \quad \forall x, y : p^\theta(y \mid x, c=0) = p^{\text{prior}}(y \mid x)$$

$$\log\frac{p^\theta(c=0 \mid x)}{p^\theta(c=1 \mid x)} = \text{KL}(p^\theta(y \mid x, c=1)\|p^{\text{prior}}(y \mid x)).$$

*Proof.* For the original KTO objective, we have:

$$\sigma\left(h_\theta(x,y) - z(x)\right) = \sigma\left(\log\frac{p^\theta(y \mid x, c=1)}{p^{\text{prior}}(y \mid x)e^{z(x)}}\right) = \frac{p^\theta(y \mid x, c=1)}{p^\theta(y \mid x, c=1) + p^{\text{prior}}(y \mid x)e^{z(x)}},$$

$$\sigma\left(-h_\theta(x,y) + z(x)\right) = 1 - \sigma\left(h_\theta(x,y) - z(x)\right).$$

And for ([13](#)), we have:

$$p^\theta(c=1 \mid x,y) = \frac{p^\theta(y \mid x, c=1)p^\theta(c=1 \mid x)}{p^\theta(y \mid x, c=1)p^\theta(c=1 \mid x) + p^{\text{prior}}(y \mid x)p^\theta(c=0 \mid x)}$$

$$= \frac{p^\theta(y \mid x, c=1)}{p^\theta(y \mid x, c=1) + p^{\text{prior}}(y \mid x)\frac{p^\theta(c=0|x)}{p^\theta(c=1|x)}}.$$

Hence the equivalence is straightforward. □

### A.2.1. COMPARISON WITH KTO

---

**Algorithm 2** KTO ([Ethayarajh et al., 2024](#))

---

1: **Input:** data $\{(x,y,c)\}_{i=1}^N$ where $c \in \{0,1\}$, a fixed reference model $p^{\text{prior}}$, trainable models $f^\theta$ initialized with $p_0$.
2: **for** every batch **do**
3:     Compute:

$$F_\theta(x,y) := \sum_t \log\left(\frac{f^\theta(y_t|x, y_{<t})}{p^{\text{prior}}(y_t|x, y_{<t})}\right).$$

4:     Estimate $z(x) := \text{KL}(f^\theta(y \mid x)\|p^{\text{prior}}(y \mid x))$.
5:     Denote $\sigma(x) := \frac{1}{1+\exp(-x)}$. Minimize loss:

$$L(x,y,c) = -c\sigma\left(F_\theta(x,y) - z(x)\right) - (1-c)\sigma\left(-F_\theta + z(x)\right).$$

6: **end for**

---

We present KTO in practice in Algorithm [2](#) using our notation for better comparison. The prior assumption on $p(c \mid x)$ does not seem to be natural and removing it yields PIPA-N. In terms of the algorithms, PIPA has the following key differences with KTO:

- PIPA has an additional learnable head $g^\theta$ to capture $p^\theta(c_t|x, y_{<t})$. Unlike the language model head, which matches the vocabulary in its output dimension, $g^\theta$ has an output dimension of only 1. Consequently, the extra parameters amount to fewer than $1\%$ of the total, resulting in no overhead in both memory and speed.

- KTO does not extend to the step-level setting, whereas PIPA seamlessly accommodates both the answer-level and step-level settings within a single framework, supported by clear theoretical guidance.

- KTO needs to additionally estimate the KL term $c(x)$ by pairing random question and answers, which adds an extra step and slows its overall process. In contrast, even with its additional learnable head, PIPA remains faster in practice.

- KTO uses the SFT model for both fixed reference model and $f^\theta$ initialization, and this is the only choice. PIPA framework allows arbitrary selection of the fixed prior model $p^{\text{prior}}$. For simplicity, we can choose $p^{\text{prior}} = f_0$ which is the same choice as KTO. But in PIPA, since the prior $p^{\text{prior}}$ is unrelated to $f^\theta$, we can also set $p^{\text{prior}}$ to be the fine-tuned version of $f_0$ on both positive and negative data to get better estimation.

- Following DPO, KTO views the log ratio $\log(f^\theta/p^{\text{prior}})$ as rewards, which is directly maximized or minimized. In PIPA-M, the ratio $\log(f^\theta g^\theta/p^{\text{prior}})$ is the log likelihood, and we need to compute $\log(1 - f^\theta g^\theta/p^{\text{prior}})$ for the negative steps, instead of things like $-\log(f^\theta g^\theta/p^{\text{prior}})$ as KTO and other works.

## B. Baseline Algorithms

**Step-DPO** The original Step-DPO (Lai et al., 2024) requires preference data generated from a tree structure and maximizes the standard DPO loss on the diverging nodes. Here, we propose a generalization of the algorithm that works with more generic paired data, without requiring a tree structure.

We are given pairwise data $(x, y^+, y^-, c^+, c^-)$ with token-level representation, where $c^+$ consists entirely of ones, while $c^-$ contains a mix of ones and zeros. First, we define

$$r_t(x, y) := \log \frac{p^\theta(y_t \mid x, y_{<t}, c_t = 1)}{p^{\text{prior}}(y_t \mid x, y_{<t})}.$$

Treating the sequences as a whole, the original DPO loss is given by

$$L_{\text{DPO}}(x, y^+, y^-, c^+, c^-) = -\log \sigma \left( \sum_t r_t(x, y^+) - \sum_t r_t(x, y^-) \right).$$

Given that the positive steps in $y^-$ can negatively impact model performance if minimized, a straightforward approach when providing step-level annotations is to exclude these steps (see e.g., Anonymous, 2025), which yields the following loss function:

$$L_0(x, y^+, y^-, c^+, c^-) = -\log \sigma \left( \sum_t r_t(x, y^+) - \sum_{t:c_t^- = 0} r_t(x, y^-) \right),$$

where we remove the positive steps in $y^-$ from the $\log \sigma()$ term. However, a potential issue arises when the number of steps varies, as the magnitude of the term inside $\sigma()$ may differ, affecting the optimization due to the $\sigma()$ function applied externally. To address this, we propose an intermediate solution between the original DPO and the above loss. Specifically, we apply the stop-gradient operation to the positive steps in $y^-$ and get $L_1$:

$$L_1(x, y^+, y^-, c^+, c^-) = -\log \sigma \left( \sum_t r_t(x, y^+) - \sum_{t:c_t^- = 0} r_t(x, y^-) - \text{sg} \left( \sum_{t:c_t^- = 1} r_t(x, y^-) \right) \right),$$

where $\text{sg}(\cdot)$ denotes stop gradient.

In essence, $L_0$ masks the implicit reward of positive steps within a negative answer in the objective, while $L_1$ masks these positive steps only during the backpropagation step when computing gradients. The loss function $L_0$ corresponds to the approach introduced in Anonymous (2025), whereas $L_1$ is equivalent to the Step-DPO formulation when using tree-structured pairwise data. Our experiments adopt $L_1$, as it demonstrates better performance.

**Step-KTO** Very recently, Step-KTO (Lin et al., 2025) introduced a loss function designed for data with step-level annotations. They partition the answer into groups corresponding to steps, where each $\sigma()$ contains only one group. For unpaired data $(x, y, c)$ represented at the token level, let $1 = s_1 < \cdots < s_K \leq T$ denote the starting tokens of all $K$ steps. Here, $c_t$ remains constant for $t \in [s_k, s_{k+1})$ for $1 \leq k \leq K$. The function $r_t(x, y)$ follows the same definition as in Step-DPO. The original KTO loss is

$$L_{\text{KTO}}(x, y, c) = c_T \sigma \left( \sum_t r_t(x, y) - z_0 \right) + (1 - c_T) \sigma \left( - \sum_t r_t(x, y) + z_0 \right).$$

The Step-KTO loss is given by

$$L_{\text{Step-KTO}}(x, y, c) = - \sum_{k=1}^{K} \left[ c_{s_k} \sigma \left( \sum_{s_k \leq t < s_{k+1}} r_t(x, y) - z_0 \right) + (1 - c_{s_k}) \sigma \left( - \sum_{s_k \leq t < s_{k+1}} r_t(x, y) + z_0 \right) \right].$$

However, our experiments revealed that Step-KTO loss does not improve performance. Inspired by the Step-DPO loss proposed earlier, we adopt the original KTO for positive answers while applying a different approach for negative answers by masking the gradient of positive steps:

$$L_1(x, y, c) = -\sigma \left( - \sum_{t:c_t=0} r_t(x, y) - \text{sg} \left( \sum_{t:c_t=1} r_t(x, y) \right) + z_0 \right).$$

The key idea is to retain the forward pass of all steps in $\sigma()$ for normalization while excluding positive steps in the backward pass to prevent their probabilities from being minimized.

