# OpenReview forum: "PIPA: Preference Alignment as Prior-Informed Statistical Estimation"
_ICML.cc/2025/Conference — ICML 2025 poster_

### Official Review · Reviewer_4Kmz · 2025-03-07

**Overall Recommendation:** 3

**Summary:**

The submission presents an approach called Prior-Informed Preference Alignment (PIPA) that accommodates paired and unpaired preference data, as well as answer and step-level annotations, for the purpose of doing offline preference alignment in language models. The approach is claimed to unify recent approaches like DPO and KTO into an RL-free probabilistic framework.

PIPA starts by assuming an (unpaired) preference dataset made of triplets $(x, y, c)$ sampled from $p^{data}(x, y, c)$, where $x$ is the instruction input, $y$ is the model response, and $c$ represents whether the response was desirable or not. It then formulates the preference learning problem as minimizing the discrepancy between the model's distribution and the data distribution subject to some prior constraint on the model's distribution. The nature of the constraint gives rise to different PIPA variants.

PIPA-M attempts to model the conditional distribution $p^\mathrm{data}(y, c \mid x)$ under the constraint that when marginalizing over $c$ the equality $p(y \mid x) = p^\mathrm{prior}(y \mid x)$ (the latter being defined through a pretrained LLM) holds. It introduces two parameterized functions $f^\theta(y \mid x)$ and $g^\theta(x)$ to represent the conditionals $p(y \mid x, c = 1)$ and $p(c = 1 \mid x)$, respectively. The constraint is enforced by construction by i) defining $p^\theta(y \mid x, c = 1) = f^\theta(y \mid x) g^\theta(x)$ and $p^\theta(y \mid x, c = 0) = p^{prior}(y \mid x) - f^\theta(y \mid x) g^\theta(x)$ and ii) enforcing that $p^\theta(y \mid x, c = 0) > 0$  via $g^\theta(x) = \min(g^\theta_0(x), p^\mathrm{prior}(y \mid x) / f^\theta(y \mid x))$.

PIPA-N instead attempts to model the conditional distribution $p^\mathrm{data}(c \mid x, y)$ under the constraint that $p^\theta(y | x, c = 0) = p^\mathrm{prior}(y \mid x)$. The parameterized $p^\theta(c = 1 \mid x, y)$ is constructed using this equality constraint and conditional probability identities, and the counterpart for $c = 0$ is obtained by noting that the two must sum up to 1 since $c$ is binary.

The step-level variants for PIPA-M and PIPA-N are defined analogously, with the difference that the functions $f^\theta$ and $g^\theta$ are defined autoregressively.

Experiments are presented for all combinations of {paired,unpaired} preferences and {answer,step}-level annotations using an unpaired preference dataset for math reasoning that includes problems from GSM8K and MATH. PIPA is compared against DPO, KTO, and IPO as well as the step-level variants of DPO and KTO and is shown to outperform competing approaches in all settings. Ablations are presented to compare PIPA-M and PIPA-N, examine the effect of several design decisions, and investigate the effect of step-level annotation.

## Update after rebuttal

The discussion with the authors addressed my concerns regarding the soundness of the proposed approach, and I feel more inclined to recommend acceptance. I have adjusted my score accordingly. I remain worried about clarity, but the authors' efforts in the discussion alleviate that to some extent.

**Claims And Evidence:**

I am not entirely convinced by the connection drawn between PIPA and DPO. Firstly, Equation 6 in Appendix A.1 represents the DPO objective only in the special case where $\beta = 1$. Secondly, and more importantly, the chosen and rejected responses are characterized as having been sampled from $p(y \mid x, c = 1)$ and $p(y \mid x, c = 0)$, respectively, but this is not how the data generating process works in practice for paired preferences: two responses are sampled, and the annotator makes a relative judgement on the two responses. This means that response $y$'s categorization into $c = 0$ or $c = 1$ is intrinsically linked to the second response $y'$ in the pair that was presented to the user: response $y$ could be bad, but so long as response $y'$ is _worse_, $y$ would be associated with $c = 1$.

**Essential References Not Discussed:**

Not that I am aware of.

**Experimental Designs Or Analyses:**

The dataset used to compare approaches makes sense in the unpaired case, but I don't think creating paired data out of correct and incorrect responses to problems is a good stand-in for a paired preference dataset (see my point about the paired/unpaired correspondence in Claims and Evidence). Given the clear dichotomy between "correct" and "incorrect" in the mathematical domain, such a procedure cannot create paired data where two responses are satisfying but one is better than the other (e.g., both are factually correct, but one is more detailed than the other), or two responses are bad, but one is worse than the other (e.g., both are factually incorrect, but one uses toxic language). It's unclear to me how PIPA's way of adapting to paired data would fare in that case in comparison to approaches specifically designed for paired data. A comparison on a paired dataset would be necessary to determine that.

I am also concerned that the model selection procedure is not sufficient to ensure that each method is presented in the best light possible. Are we sure that some methods wouldn't benefit from smaller or larger batch sizes than 256? Are we sure that some methods wouldn't benefit from a smaller or larger number of parameter updates? Why was beta chosen to be 0.1 for DPO, IPO, and Step-DPO? Do we know that this is a good hyperparameter choice for AlphaMath?

**Methods And Evaluation Criteria:**

The proposed approach makes sense in how it is constructed: having verified the derivations on my own for Sections 2.2 and 2.4, the objectives are sound and follow naturally from the definition of conditional probabilities.

One aspect of PIPA that makes less sense to me is what $g^\theta(x)$ is meant to represent. A literal interpretation would be that it is the probability of a correct answer given an instruction $x$, but that has to depend on how answers are generated from the instruction, right? If so, what generative process do we assume?

**Other Comments Or Suggestions:**

* In Section 2.2's last unnumbered equation, doesn't the optimization reduce to _maximizing_ the expected log-likelihood rather than minimizing it?
* What does the subscript 0 in $p^\mathrm{prior}_0(y \mid x)$ mean in Equation 5?

**Other Strengths And Weaknesses:**

The submission presents interesting ideas in its proposed approach.

One major weakness is its clarity. I think I now understand the ideas it tries to convey, but I had to spend a lot of time connecting the dots and navigating between sections before reaching that point. A few notable examples:

* The introduction sells the proposed approach as being applicable to both paired and unpaired data, but the preference alignment problem formulation in Section 2.1 describes the unpaired case specifically without going over the paired case. How PIPA handles paired data is only made clear in Section 3.1: "Both KTO and PIPA decouple the paired data"; "[...] we construct the paired subset by matching each correct solution to a corresponding incorrect solution from the same problem [...]". From this I infer that adapting PIPA to work on a paired dataset amounts to treating all chosen responses in the comparison pairs as having been classified as "correct/desirable" and all rejected responses as "incorrect/undesirable". This should be clarified sooner in the paper. (I also have my doubts about the validity of this adaptation; see my point in Claims and Evidence, but essentially if two responses are bad but one is less so than another, does it even make sense to say that it is "correct/desirable"?)
* The notation also contributes to the confusion: Section 2.1 talks about "estimating $p(y | x, c = 1)$", which suggests that this is the target distribution we are trying to approximate, yet the preference dataset is sampled from $p^\mathrm{data}$, and MLE usually attempts to approximate the distribution from which the empirical data is sampled. Do the authors mean that $p(y | x, c = 1)$ is the distribution with which we try to approximate $p^\mathrm{data}$? This is what Equation 1 suggests, but it should be clarified.
* The connection between PIPA-M's Equation 3 and Step 3 in Algorithm 1 is not immediately obvious. If I understand correctly, since $p^\mathrm{prior}$ does not depend on $\theta$, the derivative with respect to $\theta$ for $c = 0$ and $c = 1$ in Equation 3 are identical to that of step 3 in Algorithm 1, which is why the latter is substituted for the former to unify PIPA-M and PIPA-N in Algorithm 1. Is this correct? In any case, the substitution should be explained more clearly.

**Questions For Authors:**

1. How was the dataset partitioned for training, validation, and testing? What criterion (and what dataset split) was used to determine the best learning rate in the grid search?
2. The KTO paper also presents results on GSM8k which appear to be different from the ones presented in the submission. What explains this discrepancy?

**Relation To Broader Scientific Literature:**

In general the submission does a good job of relating to the broader scientific literature. Dumoulin et al. (2024) is cited when discussing DPO's probabilistic interpretation, but is not mentioned among the works that "[approach] alignment from a probabilistic perspective"; why is that?

**Theoretical Claims:**

I have not checked the proofs in the Appendix, but I did verify for myself that the objectives constructed in the main paper correctly follow from first principles of conditional probabilities.

---

> ### Author Rebuttal · Authors · 2025-04-01
>
> We are grateful to the reviewer for the detailed summary of the PIPA-M and PIPA-N derivations, including their extension to the step-level setting, as well as for recognizing our comprehensive experiments on math tasks.
> ## 1. Connectio between PIPA and DPO
> ### 1.1 $\beta$
> The reviewer mentions that our current PIPA-N only reduces to DPO when $\beta=1$. We emphasize that DPO with $\beta \neq 1$ directly corresponds to PIPA-N with a power prior [1] in Bayesian analysis, meaning:
> $$p(A|B) \propto p(B|A)^{\beta} p(A).$$
> Specifically, introducing $\beta$ modifies the original DPO loss as follows:
> $$\max_\theta  \quad \log\sigma\left(\beta\log \frac{p^{\theta}(y^+ | x, c = 1)p^{\text{prior}}(y^- | x)}{p^{\theta}(y^- | x, c = 1)p^{\text{prior}}(y^+ | x)}\right)
> = -\log\left(1+\left(\frac{p^{\theta}(y^- | x, c = 1)p^{\text{prior}}(y^+ | x)}{p^{\theta}(y^+ | x, c = 1)p^{\text{prior}}(y^- | x)}\right)^\beta\right).$$
> Furthermore, when incorporating the power prior $\beta$ in PIPA-N, all terms of $h_\theta(y_1,y_2,x)$ in equation (9) transform into $h_\theta^{\beta}(y_1,y_2,x)$. It is straightforward to verify that the updated PIPA-N loss with $\beta$ precisely matches the DPO loss with $\beta$.
> ### 1.2 Sampling
> We respectfully disagree with the reviewer's interpretation and emphasize that our PIPA process aligns with practice. As the reviewer noted, an answer $y$ can be classified as positive ($c=1$) or negative ($c=0$) based on its comparison with the paired answer. This implies the existence of a **non-trivial joint distribution** over $(y,c|x)$. For a high-quality response, $p(y, c=1|x)$ will be greater, while $p(y,c=0|x)$ will be higher for a less favorable response. In practice, we can model positive samples as being drawn from $p(y|x,c=1)$ and negative samples from $p(y|x,c=0)$, allowing us to apply MLE to address the problem.
> ### 1.3 Meaning of $g^{\theta}(x)$
> This represents the marginal distribution $p^{\theta}(c=1|x)$, which computes integral over all possible answers $y$. Empirically, it can be interpreted as **the difficulty of a question**. For example, if the question $x$ is "What is 2+2?", then $g^{\theta}(x)$ should be expected to be close to 1 since there's clear consensus on the correct answer. However, if the question is "What is the most beautiful color?", then $g^{\theta}(x)$ would be closer to 0.5 since this is highly subjective with no definitive answer.
> ## 2. Experiments
> ### 2.1 Paired data construction
> We respectively disagree with the reviewer’s concern about pairing correct and incorrect answer together does not help learn the preference.
> - Paired data is standard and widely used in SOTA math models training, e.g., [2]. These paired data may start from similar analysis but suddenly diverge in certain steps, which help the model learn the key steps from comparison.
> - Our experiments about win rate and KL trajectory (https://anonymous.4open.science/r/PIPA-DB7B) also clearly shows that our DPO training is effective.
> - Practical paired data construction should rely on the goal. For math, if the goal is to enhance reasoning ability, pairing correct and incorrect answers should be effective. And if the aim is to achieve an efficient and short reasoning chain, pairing a longer correct answer with a shorter correct answer is preferable. While pairing two incorrect answers is theoretically possible, it serves limited practical utility in most training scenarios. For general preference alignment, we also don’t think pairing two bad answers (bad versus worse) is helpful. We believe robust filtering mechanisms should be implemented to avoid such non-productive training signals.
> ### 2.2 Hyperparameters
> Our hyperparameter choice is based on rigorous selection to ensure fair comparison across all methods.
> - **Batch size**: Since PIPA only modifies the loss function compared to baselines, **a fair comparison would be keeping the batch size and training steps to be the same across all algorithms.** And 256 is a standard choice  for datasets containing tens of thousands of examples.
> - **$\beta$**: See the results in Section 1 of response to 3B8U.
> ### 2.3 Data split
> We use standard benchmarks (GSM8K, MATH) for testing and Alphamath for training (10% for validation). LR search follows standard paradigm using train loss and validation loss.
> ### 2.4 Results differences
> They are from differences in models and training data.
> ## 3. Writing Improvements and Clarifications
> We thank reviewers for their careful reading and suggestions. We will:
> - Add Dumoulin et al. (2024) to the related work section.
> - Clarify PIPA's approach for paired data earlier.
> - Clarify the training objective explanation.
> - Clarify substitutions in Equation 3 and step 3.
> - Change minimizing to maximizing.
> - Remove the 0 in Equation 5.
>
> [1] Ibrahim J G, et al. The power prior: theory and applications[J]. Statistics in medicine, 2015.
>
> [2] Xiong, Wei, et al. “Building Math Agents with Multi-Turn Iterative Preference Learning.” ICLR 2025.

---

> > ### Comment · Reviewer_4Kmz · 2025-04-04
> >
> > Thank you. Below are my comments following your response.
> >
> > **1.1** I am satisfied with your explanation. It's important for clarity that this is also made explicit in the paper.
> >
> > **1.2** I think I'm starting to see the point you are making. Please let me know if this is correct: the joint $p(y, c | x)$ can be thought of as implicitly marginalizing over all possible $y'$ alternatives and reflects the marginal probability that $y$ would be preferred over any $y'$. If so, this is reminiscent of how Munos et al. (2024) define the preference between two distributions conditioned on a state $x$.
> >
> > **1.3** The interpretation makes sense; however I'm still not clear on what distribution we marginalize over when computing the integral over all possible answers $y$. Can you elaborate?
> >
> > **2.1** I am satisfied with the explanation in the context of math problems, but I disagree that pairing two bad answers (bad versus worse) is unhelpful in general preference alignment. The precise reason why one bad answer is better than a worse answer can still yield some useful signal: to reuse the example in my review, if two answers are factually incorrect but one uses toxic language, we can still get a useful signal on toxicity. I think this disagreement is illustrative of a broader worry I have with the proposed approach's general applicability, which is that it works best when there is a clear dichotomy in preferable and non-preferable answers. Note that this by itself is not a fatal flaw in the submission (I recognize that even in this more restricted scope valuable contributions can be made), but it is an interrogation point that I could see people having when deciding whether to adopt PIPA for their own preference learning problem.
> >
> > **2.2** I remain concerned: I don't agree that keeping the batch size and training steps to be the same across all algorithms is the fair thing to do. A modified loss function could mean a completely different loss landscape requiring more or less steps to converge, for instance.
> >
> > **2.3** I am satisfied with the clarification. Please make sure to mention those details in the paper.
> >
> > **2.4** Can the authors elaborate on the differences? Isn't KTO the same between the paper that proposes it and this submission?
> >
> > **3** I am satisfied with the authors' response.
> >
> > **References**
> >
> > Munos, R., Valko, M., Calandriello, D., Azar, M. G., Rowland, M., Guo, Z. D., ... & Piot, B. (2024). Nash learning from human feedback. In Proceedings of the International Conference on Machine Learning.

---

> > > ### Author Response · Authors · 2025-04-05
> > >
> > > Thank the reviewer for your thorough reading of our rebuttal and for replying to each point. We are glad that we address many of your concerns.
> > >
> > > **1.1** Thank you for the satisfaction, we will include this in the paper.
> > >
> > > **1.2** Thank you for agreeing with our explanation. Yes your understanding is correct. We will clarify this further in the paper and cite Munos et al. (2024).
> > >
> > > **1.3** Thank you for agreeing with our interpretation. Theoretically, the integral is taken over all possible answers $y$, that is, $p^\theta(c=1|x) = \int p^\theta(c=1|x,y) p^\theta(y|x) dy$. In our PIPA framework, we approximate $p^\theta(c=1|x)$ directly using a neural network that depends on $x$.
> > >
> > > **2.1** Thank you for your positive response to our explanation regarding pairwise data in the math context, and for recognizing the effectiveness of PIPA in good-bad pair scenarios. Regarding cases like good-better and bad-worse:
> > >
> > > - As you mentioned in **1.2**, the PIPA framework does cover situations where labels are derived after comparison. In this paper, we primarily focus on the KTO-style PIPA for decoupled data. However, in scenarios where coupled data is necessary (as in good-better or bad-worse comparisons), **a DPO-style PIPA can also be applied**. In particular, Equation (9) in our paper reduces exactly to DPO when assuming a prior of $p(c=1|x) = 0.5$. By removing this prior, we arrive at the paired version of PIPA-N, which introduces **an additional margin term** $\frac{p^\theta(c=1|x)}{p^{\theta}(c=0|x)}$ to the DPO loss. We present experimental results for this variant and show the advantage in Section 2 of our response to 3B8U.
> > >
> > > - Empirically, PIPA only modifies the loss functions in DPO/KTO etc. Therefore, it remains fully compatible with any dataset used by those approaches, without imposing further constraints on data selection.
> > >
> > > - The question of whether bad-worse pairs are useful is beyond the scope of our work and pertains more broadly to the field of preference alignment. Here we agree with your point that, since methods like DPO and paired PIPA optimize for preference gaps, comparisons between bad and worse can also provide useful information for the model.
> > >
> > >
> > > **2.2** We evaluate batch sizes of {64, 256, 1024} for DPO, IPO, and PIPA. Across all batch sizes, PIPA consistently outperforms the other methods, and the performance trends remain similar. As shown in the table, PIPA achieves higher accuracy both when comparing results at each batch size and when considering the best-performing batch size for each algorithm.
> > >
> > > | Algorithm | Batch Size | GSM8K Accuracy | MATH Accuracy |
> > > |-----------|------------|----------------|---------------|
> > > | DPO       | 64         | 69.60          | 45.48         |
> > > | DPO       | 256        | 68.39          | 46.94         |
> > > | DPO       | 1024       | 68.69          | 47.14         |
> > > | IPO       | 64         | 71.42          | 47.58         |
> > > | IPO       | 256        | 69.14          | 46.94         |
> > > | IPO       | 1024       | 68.69          | 46.56         |
> > > | PIPA      | 64         | **81.35**      | **52.34**     |
> > > | PIPA      | 256        | **79.08**      | **50.82**     |
> > > | PIPA      | 1024       | **75.74**          | **47.60**         |
> > >
> > >
> > > **2.3** Thank you for the satisfaction, we will include this information in the paper.
> > >
> > > **2.4** PIPA, DPO, and KTO differ only in their loss functions, while GSM8K serves purely as a testing benchmark. Therefore, differences in pre-trained models, training data, and evaluation strategies can naturally lead to variations in results.
> > >
> > > Specifically, the KTO paper fine-tunes general-purpose models like LLaMA-3-8B and Qwen-2.5-3B-Instruct on the UltraFeedback dataset, and evaluates using an 8-shot setting.
> > > In contrast, our experiments fine-tune stronger models—Deepseek-Math-7B-Instruct and Qwen-2.5-Math-7B-Instruct on the high-quality Alphamath dataset. We also adopt a zero-shot evaluation strategy. By leveraging more capable models and domain-specific data, **we aim to establish stronger baselines and narrow the room for improvement, making any gains achieved by better loss functions even more impressive.**
> > > Moreover, zero-shot evaluation helps us isolate the effect of the loss function itself, avoiding confounding factors such as the randomness or relevance of few-shot examples. Overall, our setup is **carefully designed to provide a clean, fair and challenging benchmark for comparing different algorithms.**
> > >
> > >
> > > **3** Thank you for the satisfaction.

---

### Official Review · Reviewer_Zrno · 2025-03-14

**Overall Recommendation:** 3

**Summary:**

The paper proposes a prior-oriented perspective on effectively leveraging negative samples in preference learning, given that MLE (SFT) is the optimal solution with positive samples. Within this perspective, DPO and KTO are unified by their use of different priors and loss functions for negative samples.

Building on this framework, the authors introduce PIPA-N and PIPA-M as two design approaches, which naturally incorporate learning a value model for token-level valuation. Experimental results on paired/non-paired data, as well as outcome-level and step-level labels, demonstrate the effectiveness of the proposed methods

---

Post-rebuttal update:

I thank the authors for their rebuttal, which addresses most of my concerns. I continue to recommend acceptance of this paper, as I find the proposed framework to be a valuable contribution, and the experimental results adequately support the claims. My remaining concern lies primarily in the clarity and writing quality of the paper. With significant improvements in presentation, I would give a score of 4. As it stands, please interpret my final score as a 3.5, if such granularity is permissible.

**Claims And Evidence:**

The claims are generally well-supported. However, there are concerns regarding which specific design choices contribute to the observed experimental results and the comparison to other baselines. Please see the #2 and #4 question on the experimental analysis.

**Essential References Not Discussed:**

To the reviewer’s knowledge, all significant references have been appropriately discussed.

**Experimental Designs Or Analyses:**

1. Table 2 lacks a clear explanation of the experimental settings. The reviewer has to guess that this is the DeepSeek model and the step-wise approach for unpaired data.

2. Table 2 raises a question about the source of performance improvement. Does the observed improvement primarily stem from the learned token-level values, rather than modifications in the PIPA formula? The results with a fixed $p^{\theta}(c_t|x,y_t)$ appear similar to step-KTO, suggesting that the gains might not be attributed to changes in the formula itself.

3. Figure 3: On which dataset was the likelihood calculated?

4. The PIPA methods apply token-level rewards, whereas the baseline methods rely on outcome-level or step-level rewards. How does PIPA compare to other approaches that also utilize token-level rewards within the DPO framework (e.g., RTO, TDPO, OREO)?

**Methods And Evaluation Criteria:**

The methods are generally well-founded. However, the reviewer has the following question on design choice in PIPA-N and the token-level reward:

In PIPA-N, why is learning $p^{\theta}(c=1|x)$ considered a simpler and more natural design compared to DPO and KTO? A more detailed explanation would help clarify the advantages of this approach over existing methods.

The token-level reward is defined by the equation at line 235. If $c$ epresents correctness, then the definition is well-defined. However, when $c$ represents preference, what does it mean in the context of $y_{<t}$?

**Other Comments Or Suggestions:**

1. Line 152 (left): The sentence is not fully clear. Should "Paramter" be "parameterize"? What does this sentence meant to say?

2. The notation $p^{\theta}$ and $g^{\theta}$  both include $\theta$, which is confusing before reaching Section 2.6.

**Other Strengths And Weaknesses:**

The prior-oriented perspective is novel and effectively illustrates the distinction between SFT and offline preference alignment methods.

The definition and introduction of the token-level reward is novel.

**Questions For Authors:**

After reading the paper, the reviewer feels that the authors attempt to cover too much content within the main text. The paper would benefit from moving some ablation studies to the appendix while expanding the explanations in Sections 2.2, 2.3, and 2.4 for better clarity.

Additionally, could the proposed framework also encompass other offline preference learning methods discussed in the related work section?

**Relation To Broader Scientific Literature:**

The paper aims to unify offline preference alignment algorithms from a prior-oriented perspective. It introduces a general framework that encompasses DPO and KTO, simplifying them into practical algorithms that outperform previous methods.

Additionally, the paper provides intuitions on the distinction between SFT and preference alignment, framing it as the incorporation of a prior on negative responses.

**Theoretical Claims:**

It would be helpful to provide a more detailed explanation of the reduction after Theorem 2.1, particularly why $g_{0}^{\theta}$ disappears from the objective. The reviewer understands that $g^{\theta}$ is introduced to satisfy the constraint and is inherently not a variable to optimize, but this could be made clearer to the reader.

---

> ### Author Rebuttal · Authors · 2025-04-01
>
> We appreciate the reviewer for acknowledging our novel well-supported framework and comprehensive experiments in math. We will improve writing to make presentation more clear.
> ## 1. Why learnable $p^\theta(c=1|x)$ is better?
> Unlike DPO (which uses a simple 0.5 prior) and KTO (which uses a complex KL-based prior), our approach makes this term learnable—a more elegant solution. As explained in our response to reviewer 4Kmz (section 1.3), this term can be interpreted as problem difficulty and is better learned during training rather than using a predetermined fixed value.
> ## 2. Meaning of $y_{<t}$
> We agree that conditional independence only works for math-like tasks with clear good/bad examples. However, for scenarios involving good/better comparisons you mentioned, a DPO-style paired loss function is necessary rather than the current KTO-style unpaired approach presented in our work, and it also means there is no step-level signals due to the uncertainty.
> To get the new algorithm in the new setting, we can follow Theorem A.1's derivation and design a paired PIPA variant (not explored in our current version which focuses on KTO-style unpaired PIPA). Starting from equation (9), we can expand terms in $h_\theta$ autoregressively while directly learning $p^{\theta}(c=1|x)$.
> This deriviation does not need token-level labels, avoiding the concerns you mentioned.
> ## 3. Writing improvements and clarifications
> Thank the reviewer for the careful reading and suggestions to improve our work. We will
> - Explain $g_0^\theta$ more clearly in Thm 2.1.
> - Clarify the setting in Table 2.
> - Change Parameter to parameterize in Line 152.
> - Make notation with $\theta$ more clear.
> - Reorganize the sections and move some experiments to appendix to make presentation more clear.
>
> Dataset used in Figure 3: The likelihood is computed on Alphamath during training.
> ## 4. Experiments
> ### 4.1 Source of advantages
> Our approach gains benefits from both formula and token-level rewards—these components work effectively and neither functions effectively without the other. Importantly, integrating token-level rewards into DPO/KTO frameworks is non-trivial and it’s not straightforward to design an algorithm that only employs token level rewards without appropriate changes to the formula.
> ### 4.2 Baselines
> We don't include comparisons with RTO and OREO because they require an additional learnable model, unlike our approach which only needs an extra prediction head. This makes our method more directly comparable to the original DPO and KTO implementations with similar computational costs. We add the comparison with TDPO below using Deepseek model, and PIPA is better. More comparison will be included later due to the limited response period.
>
> | Algorithm | Learning Rate | $\beta$ | Batch Size | GSM8K Accuracy | MATH Accuracy |
> |-----------|---------------|------|------------|----------------|---------------|
> | TDPO | 1e-5 | 1.0 | 256 | 76.15 | 48.84 |
> | PIPA-M | 5e-5 | NA | 256 | 79.08 | 50.82 |
>
> ## 5. Encompassing other methods
> The answer is yes. In paired PIPA, equation (9) in Theorem A.1 can be formulated as DPO plus a margin term structured as $\log\frac{p^\theta(c=0|x)}{p^\theta(c=1|x)}$, which enables us to cover margin-based approaches similar to SimPO. Additionally, by introducing a power prior into PIPA, we can encompass DPO variants with $\beta\neq 1$. See section 1.1 in the response to reviewer 4Kmz for details.
> ## 6. Why math tasks?
> Here we provide further explanation on our choices of using math tasks in this paper.
> ### 6.1 Math is a natural fit
> Given that our PIPA framework provides key advantages—such as removing the requirement for paired data and naturally extending to step-wise labels—we prioritize math tasks as the optimal testbed for validation. This enables us to benchmark PIPA against general baselines like DPO and more specialized approaches such as step-DPO and step-KTO, which are designed to tackle these challenges.
> ### 6.2 Easy generalization to other tasks
> We appreciate the suggestion to expand the scope of our study. Our PIPA framework is applicable to general preference tasks, particularly through the paired PIPA formulation outlined in Theorem A.1 of our paper. Preliminary results are presented in Table 4 in response to Review 3B8U. Further exploration about paired PIPA and other tasks will be explored more in future work.
> ### 6.3 The importance of paired data in math
> While some math benchmarks use rule-based rewards, a preference signal that distinguishes correct from incorrect answers remains crucial. It helps the model recognize and learn from flawed reasoning or incorrect steps within the same problem [1, 2, 3…]. The impact of paired data is more pronounced when high-quality paired data is available. Thus, when designing alignment algorithms, it is essential to consider both paired and unpaired data. PIPA serves as a promising step in exploring this direction. See more discussions in Section 4.2 in the reply to 4Kmz.

---

### Official Review · Reviewer_591D · 2025-03-16

**Overall Recommendation:** 3

**Summary:**

**Post-rebuttal Update**: I thank the authors for their detailed explanation on the unified framework. I have adjusted my score to 3.

---

The paper presents a novel framework called Prior-Informed Preference Alignment (PIPA), which unifies various preference optimization objectives for language model alignment. The authors propose two variants, PIPA-M and PIPA-N, both formulated as constrained maximum likelihood estimation (MLE) problems with prior constraints. The PIPA framework effectively generalizes existing methods like Direct Preference Optimization (DPO) and Kahneman-Tversky Optimization (KTO) by incorporating prior-informed constraints. Empirical results on GSM8K and MATH benchmarks demonstrate that PIPA achieves consistent performance gains without additional computational costs.

**Claims And Evidence:**

The authors claim that PIPA unifies multiple preference optimization approaches, generalizes existing methods, and improves performance. Theoretical analysis justifies the framework's design, while extensive experiments across answer-level and step-level tasks validate these claims.

However, the observed performance difference between PIPA-M and PIPA-N lacks a consistent explanation. Neither can consistently surpass the other. This raises questions about the underlying motivations for these two variants.

**Essential References Not Discussed:**

None that I am aware of.

**Experimental Designs Or Analyses:**

The experiments are comprehensive and address key aspects of preference optimization across different data types (paired/unpaired and answer/step-level). However, the comparison to DPO, IPO, and KTO in the step-level setting may be somewhat unfair since these methods are not inherently designed for multi-step math/reasoning tasks. A comparison to reinforcement learning (RL) methods would be more appropriate given the rule-based reward signals available for math tasks.

**Methods And Evaluation Criteria:**

The proposed methods are sound and align well with established evaluation criteria for preference optimization in language models. The authors provide a clear derivation of their approach and conduct comprehensive experiments on relevant benchmarks (GSM8K and MATH).

The choice to test on math benchmarks may limit the broader applicability of the proposed method. I wonder why the authors chose to test on math benchmarks, where a rule-based reward signal is available and there is no pressing need to construct preference signals. Alternatively, have the authors tested PIPA on alignment tasks, where the pairwise preference makes more sense?

**Other Comments Or Suggestions:**

typos:
1. Line 140, left column: This is by definition is...
2. Line 151, left column: we are going to parameter (parameterize?)

**Other Strengths And Weaknesses:**

Strengths:

+ The proposed framework has the potential to offer a unified understanding of preference optimization techniques, which might be of independent interest.

Weaknesses:

+ The empirical distinction between PIPA-M and PIPA-N lacks a strong theoretical grounding or unifying motivation.

+ Testing only on math benchmarks underutilizes the potential impact of PIPA on alignment tasks with human preferences.

**Questions For Authors:**

1. Can the authors provide a clearer explanation for the differing motivations behind PIPA-M and PIPA-N? How can these methods be unified under a common perspective?

2.  There is an impression that PIPA-M and PIPA-N are quite different algorithms, and empirically there is no consistent advantage for one to be better than the other. Is there any more unified motivation so that PIPA-M and PIPA-N are motivated from the same starting point? Specifically, eqn (2) and (5) are not quite related to each other, except that they can fall under (1) as a constrained optimization problem.

3. The answer-level setting is reasonable to me. I wonder why the authors chose to test on math benchmarks, where a rule-based reward signal is available and there is no pressing need to construct preference signals. Alternatively, have the authors tested PIPA on alignment tasks, where the pairwise preference makes more sense?

4. For the step-level setting, the comparison to DPO, IPO or KTO might be slightly unfair as they are not originally designed for math or reasoning tasks. Instead, given the rule-based reward, comparison to RL algorithms is more suitable, suppose a process reward signal is available. And even if there is no process reward signal, the RL algorithms can still optimize on the rule-based outcome reward without any computation overhead.

**Relation To Broader Scientific Literature:**

None that I am aware of.

**Theoretical Claims:**

The derivations appear correct to me.

---

> ### Author Rebuttal · Authors · 2025-04-01
>
> We appreciate the reviewer for acknowledging our innovative framework and thorough experiments for mathematical tasks.
>
> ## 1. Unified derivation of PIPA-M and PIPA-N
> Both PIPA-M and PIPA-N derive from the same MLE target, differing only in their prior assumptions. For data (x, y, c) where:
> - x: question
> - y: answer
> - c: label (1=pos, 0=neg)
>
> We want to use MLE to solve $p^{\theta}(c | x,y)$. By Bayes’ rule:
> $$p^{\\theta}(c=1 | x, y) = \\frac{p^{\\theta}(y|x, c=1) \\cdot p^{\\theta}(c=1|x)}{p^{\\theta}(y|x)} \quad \\text{and} \quad p^{\\theta}(c=0 |x, y) = 1-p^{\\theta}(c=1 | x,y)$$
> Now how to parameterize the 3 terms in $p^{\\theta}(c=1 | x, y)$? In both PIPA-M and N, the 2 terms in numerator are learnable:
> - $p^{\theta}(y|x, c=1)$ corresponds to $f^{\theta}(y|x)$ in our paper
> - $p^{\theta}(c=1|x)$ corresponds to $g^{\theta}(x)$ in our paper
>
> The key difference lies in how we handle the denominator $p^{\theta}(y|x)$:
> - *PIPA-M*: Uses a fixed prior for the margin distribution: $p^{\theta}(y|x) = p^{prior}(y|x)$
> - *PIPA-N*: Uses a fixed prior for the negative conditional distribution. So we expand the marginal distribution to conditional distribution:
> $$\\begin{align}
> p^{\\theta}(y|x) &= p^{\\theta}(y|x,c=1)p^{\\theta}(c=1|x) + p^{\\theta}(y|x,c=0)p^{\\theta}(c=0|x) \\\\
> &= f^{\\theta}(y|x) \\cdot g^{\\theta}(x) + p^{prior}(y|x) \\cdot (1-g^{\\theta}(x))
> \\end{align}$$
>
> This concise derivation illustrates the relationship between PIPA-M and PIPA-N. **The reviewer questions the relationship between (2) and (5). We point out here that they are the same problem with different priors.**
> Equation (5) for PIPA-N is exactly the MLE above. The equivalence between the above MLE for PIPA-M and the KL objective (2) is from the marginal prior and is demonstrated in Lines 144-147 of our paper. In our next revision, we will improve the presentation to provide a more unified description of both approaches.
> ## 2. Empirical understanding about performance difference
> As mentioned in Lines 318-329, our empirical findings suggest different optimal scenarios for each approach. Below, we provide additional understanding for these observations:
> ### 2.1 From the prior
> - **PIPA-N performs better for self-alignment**: When data is generated on-policy from previous model versions, our goal is self-improvement. In this context, it's reasonable to assume the current model $p^{prior}$ better resembles the negative-condition distribution.
> - **PIPA-M performs better under distribution shift**: When using data from external sources, the distribution shift makes it more appropriate to use the current model as a prior for the overall distribution rather than specifically for the negative class, which would be an aggressive assumption.
> ### 2.2 From the loss
> A deeper analysis of the loss functions of positive samples also show the differences:
>
> **PIPA-M Loss:**
> $$-\\log\\frac{p^{\\theta}(y|x,c=1)p^{\\theta}(c=1|x)}{p^{prior}(y|x)}$$
>
> This effectively maximizes $p^{\\theta}(y|x,c=1)$ and $p^{\\theta}(c=1|x)$ directly.
>
> **PIPA-N Loss:**
> $$-\\log \left(p^{\\theta}(y|x,c=1)p^{\\theta}(c=1|x)\right) + \\log \left(p^{\\theta}(y|x,c=1)p^{\\theta}(c=1|x) + p^{prior}(y|x)(1-p^{\\theta}(c=1|x)\right)$$
>
> PIPA-N incorporates an **additional regularization term** that prevents $p^{\\theta}(y|x,c=1)$ from overfitting to the training data. This regularization helps the model avoid getting stuck at the checkpoint that generated correct but suboptimal examples, which is particularly beneficial for self-improvement scenarios.
> ## 3. General preference tasks
> Please see Table 4, in Seciton 2 of the response to 3B8U.
> ## 4. Why math tasks?
> See Section 6 in the reply to Zrno.
> ## 5. Why use DPO/KTO for baselines?
> DPO/KTO and their variants (iterative, stepwise, etc.) are widely acknowledged as **the primary—and often the only—baselines** in most alignment research on mathematical reasoning [1,2,3,4…]. We respectfully disagree with the reviewer’s suggestion to use RL methods like PPO/GRPO as baselines. These are **on-policy** algorithms, whereas the original DPO/KTO and our PIPA are all **off-policy** methods using a fixed dataset that differ only in their loss functions. Comparing off-policy methods with RL approaches would be inherently unfair. When comparing DPO/KTO in our work, we keep everything the same except the loss functions. A fair comparison with RL would require considering iterative/online DPO and developing an online version of PIPA, which extends far beyond the scope of this study.
>
> ## 6.Typos
> We will fix the typos in Line 140, 151.
>
> [1] Pang, Richard Yuanzhe, et al. "Iterative reasoning preference optimization." NeurIPS 2023.
>
> [2] Chen, Guoxin, et al. “Step-level Value Preference Optimization for Mathematical Reasoning.” EMNLP 2024.
>
> [3] Wang, Huaijie, et al. "Offline Reinforcement Learning for LLM Multi-Step Reasoning."
>
> [4] Xiong, Wei, et al. “Building Math Agents with Multi-Turn Iterative Preference Learning.” ICLR 2025.

---

### Official Review · Reviewer_3B8U · 2025-03-21

**Overall Recommendation:** 3

**Summary:**

The paper presents Prior-Informed Preference Alignment (PIPA), a unifying probabilistic framework for offline preference tuning of language models. It views alignment as a maximum likelihood estimation with constraints that tie the “correct” and “incorrect” output distributions to a reference prior. Within this perspective, the authors explain how existing algorithms like DPO and KTO are special cases with different ways of imposing prior constraints. They also propose two PIPA variants (PIPA-M and PIPA-N), each incorporating the prior differently, and how they can handle both answer-level and step-level annotations. Experimental evaluations on GSM8K and MATH confirm that PIPA consistently outperforms baselines under multiple conditions while retaining a simple, efficient training procedure.

**Claims And Evidence:**

Overall, most of the paper’s claims are supported by evidence in the form of clear theoretical derivations (showing how DPO and KTO emerge as special cases of PIPA). However, one possible limitation is the scope of experimentation: all empirical results come from math-focused datasets (GSM8K and MATH). While these are strong tasks for stepwise feedback, it’s less clear how PIPA behaves for broader preference-alignment scenarios.

**Essential References Not Discussed:**

In line 421, the authors discuss previous works showing that DPO decreases the probability of positive samples during training. They, however, left out one work that also shows this problem (“Provably Mitigating Overoptimization in RLHF: Your SFT Loss is Implicitly an Adversarial Regularizer” https://arxiv.org/abs/2405.16436).

**Experimental Designs Or Analyses:**

From an experimental design perspective, one area that could be strengthened is hyperparameter validation. While the paper mentions a grid search, it isn’t entirely clear whether the proposed methods and baselines are treated equally or whether the chosen hyperparameters reflect their optimal settings. From my experience, simple hyperparameter tuning can make it seem like a proposed method shows better performance than the baseline.
A helpful way to address potential “hyperparameter hacking” is to plot a “GT win rate” (or accuracy/performance) against the KL-divergence across a dense grid of hyperparameter values for both the new methods and the baselines (similar to approaches in other RLHF or offline alignment work (https://arxiv.org/abs/2406.02900) which uses a similar plot for observing overoptimization). Such plots would demonstrate (1) whether the selected hyperparameters lie near a reasonable optimum and (2) that the same thoroughness is applied to all methods. Without this, one might worry that the authors’ methods have undergone more meticulous tuning than the baselines, potentially skewing the comparison.

**Methods And Evaluation Criteria:**

While the math-domain experiments are systematic and informative, a key limitation is that both the tasks (GSM8K, MATH) and the models (Deepseek-Math-7B, Qwen2.5-Math-7B) are math-focused, which may not fully generalize to broader open-ended preference scenarios (e.g., creative writing, summarization, or subjective opinion tasks). In particular, it is unclear whether step-level preference alignment would improve performance on non-math benchmarks where correctness is less binary. For example, I would suggest evaluating the proposed methods on widely used preference alignment benchmarks, such as AlpacaEval and ArenaHard.
Moreover, while the authors compare PIPA against DPO and KTO variants, they omit more recent or higher-performing preference-alignment methods like SimPO, which reportedly achieves strong results on open-ended generation tasks.

**Other Comments Or Suggestions:**

It would be nice to see how the recently proposed SimPO, which shows significant performance gain compared to DPO or other DPO variants, fits into the proposed framework.
Moreover, as discussed earlier in my review under [Experimental Designs Or Analyses], it is essential to perform experiments on non-math-related tasks and validate the choice of hyperparameter by showing the Performance vs. KL-Divergence curve.

**Other Strengths And Weaknesses:**

The most original aspect is the unified viewpoint that treats offline preference alignment as a constrained MLE problem. This clarifies perspectives, bridging various existing methods—DPO, KTO, Step-DPO, etc.—under one probabilistic framework. The theoretical consolidation is meaningful: clarifying how each alignment technique differs only by the choice of prior constraint (marginal vs. negative) can help practitioners see how new or existing approaches fit in.

**Questions For Authors:**

1.	As discussed earlier, could the authors show the Performance vs. KL-Divergence to validate their hyperparameter searching?
2.	Could the authors show their method on non-math related tasks (Alpaca Eval, Arena Hard)
3.	Also, could the authors compare their method to SimPO?

**Relation To Broader Scientific Literature:**

A number of recent works (e.g., DPO, KTO) already eschew traditional RL algorithms in favor of more direct, offline approaches. However, these methods typically target either paired preference data (DPO) or unpaired data (KTO) without a unifying outlook. The paper’s “prior-informed” lens systematically shows how such offline algorithms can be viewed as special cases of one probabilistic formulation.

**Theoretical Claims:**

The main theoretical claims center on
1.	Showing that DPO and KTO emerges as special cases by imposing different prior constraints on the model distribution (theorem A.1, theorem A.2)
2.	Constructing a parameterization that satisfies a marginal or negative-prior constraint (theorem 2.1)
The derivations are consistent with standard probabilistic modeling arguments, and there do not appear to be any major errors

---

> ### Author Rebuttal · Authors · 2025-04-01
>
> We appreciate the reviewer for the recognition of our theoretical contribution in unifying existing preference alignment methods under a single probabilistic framework, and our experiments demonstrating PIPA's effectiveness on math reasoning tasks with stepwise feedback.
>
> ## 1. Hyperparameter validation
>
> **Validation via learning trajectory**
> Following the paper referenced by the reviewer, we plot the trajectory of the forward KL divergence and the win rate computed by the implicit reward during training on Alphamath with Deepseek (https://anonymous.4open.science/r/PIPA-DB7B). Both metrics indicate that DPO effectively learns throughout the training process without exhibiting the over-parameterization issues highlighted in the referenced paper.
>
> **Validation via additional results**
> Our hyperparameters were carefully selected through rigorous search, ensuring that they were not cherry-picked to favor our results. To illustrate this, we present additional results for batch size, $\beta$, and learning rate—three key hyperparameters in DPO. The current settings in the paper are (256, 0.1, 5e-7). We here find a slightly larger $\beta$ than 0.1 allows for a higher learning rate and improves performance, leading to better results. However, DPO still falls short of PIPA. Specifically:
>
> - **Batch size:** Consistent with previous studies, we use a fixed batch size across all algorithms to ensure a fair comparison by maintaining the same number of updates. We evaluate batch sizes of {64, 256, 1024} for DPO, IPO, and PIPA, and the results consistently show PIPA’s advantage at all batch sizes.  We use 5e-7 for DPO and IPO here because larger LR will crash.(https://anonymous.4open.science/r/PIPA-DB7B/table1.md)
>
> - **$\beta$:** Initially, we set $\beta$ to 0.1 following the original Alphamath paper [1]. We then explored values from {0.5, 1.0, 2.0, 5.0} and found that $\beta$ = 1.0 yielded the best results. However, DPO still lags behind PIPA. (https://anonymous.4open.science/r/PIPA-DB7B/table2.md)
>
>
> - **Learning rate:** With optimal batch size and $\beta$, we try learning rates from {5e-7, 1e-6, 5e-6, 1e-5, 5e-5}. We find increasing $\beta$ to 1.0 allows for a higher optimal learning rate of 1e-5. This adjustment improves DPO's performance on the GSM8K dataset, reaching 0.74 and narrowing the gap with PIPA to 5%. However, it does not lead to any improvements on the more challenging MATH dataset. (https://anonymous.4open.science/r/PIPA-DB7B/table3.md)
>
>
> ## 2. Non-math preference tasks
> We employed the paired version of PIPA-N as outlined in Theorem A.1, with further algorithmic details provided in Section 2 of response to Reviewer Zrno. Following the setup proposed in SimPO, we conducted comparisons between DPO and SimPO by fine-tuning Llama-3-8B-instruct on the Ultrafeedback dataset. We reproduced the DPO and SimPO baselines in our codebase using OpenRLHF, maintaining identical hyperparameters from the SimPO paper (e.g., batch size, epochs, $\beta$, $\gamma$), except for using LoRA (r=64, $\alpha$=16) with a higher learning rate, as the original lower learning rate is ineffective with LoRA. Our reproduction produced results matching those reported by SimPO.
> For our PIPA-N, current results are better than SimPO. It’s noteworthy that due to the limited rebuttal period, we did not have the resources to thoroughly optimize the hyperparameters for PIPA-N. In contrast, the SimPO results benefited from careful tuning (e.g., using $\beta$=2.5 and $\gamma$=1.375 for Llama-3-8B-instruct). This suggests that PIPA-N has the potential for further improvement with more extensive parameter tuning. More results will be provided later given the limited response period.
>
> | Algorithm | AE2LC | AE2WR|  AH |
> |-----------|-------|---------|----------|
> | DPO | 41.17 | 36.25 | 30.47 |
> | SimPO | 43.28 | 39.86 | 32.49 |
> | PIPA-N | **44.79** | **40.53** | **33.08** |
> *Table 4: Non-math tasks.*
>
> ## 3. Comparison to SimPO
> Table 2 shows that SimPO is similar to DPO in Alphamath which lags behind PIPA, and Table 4 shows that PIPA-N is also better than SimPO.
>
> ## 4. Missing citation
> We will cite the paper by Zhihan Liu et al. in Line 421.
>
> [1] Chen Guoxin, et al. AlphaMath Almost Zero: Process Supervision without Process, NeurIPS 2024.

---

### Decision · Program_Chairs · 2025-05-01

**Decision:**

Accept (poster)

**Comment:**

The paper introduces Prior-Informed Preference Alignment (PIPA), a probabilistic framework for offline preference tuning of language models. PIPA is an intriguing concept, and most claims are well-supported by clear theoretical derivations. Experimental evaluations on GSM8K and MATH show that PIPA consistently outperforms baseline models. Overall, this is a good work. To enhance the final paper, the authors should improve the presentation clarity, include results beyond math benchmarks, and provide more discussion on the distinctions between the two model variants.